# A New Neural Kernel Regime:
# The Inductive Bias of Multi-Task Learning

**Julia Nakhleh**
Department of Computer Science
University of Wisconsin-Madison
Madison, WI
jnakhleh@wisc.edu

**Joseph Shenouda**
Department of Electrical and Computer Engineering
University of Wisconsin-Madison
Madison, WI
jshenouda@wisc.edu

**Robert D. Nowak**
Department of Electrical and Computer Engineering
University of Wisconsin-Madison
Madison, WI
rdnowak@wisc.edu

## Abstract

This paper studies the properties of solutions to multi-task shallow ReLU neural network learning problems, wherein the network is trained to fit a dataset with minimal sum of squared weights. Remarkably, the solutions learned for each individual task resemble those obtained by solving a kernel regression problem, revealing a novel connection between neural networks and kernel methods. It is known that single-task neural network learning problems are equivalent to a minimum norm interpolation problem in a non-Hilbertian Banach space, and that the solutions of such problems are generally non-unique. In contrast, we prove that the solutions to univariate-input, multi-task neural network interpolation problems are almost always unique, and coincide with the solution to a minimum-norm interpolation problem in a Sobolev (Reproducing Kernel) Hilbert Space. We also demonstrate a similar phenomenon in the multivariate-input case; specifically, we show that neural network learning problems with large numbers of tasks are approximately equivalent to an $\ell^2$ (Hilbert space) minimization problem over a fixed kernel determined by the optimal neurons.

## 1 Introduction

This paper characterizes the functions learned by multi-output shallow ReLU neural networks trained with weight decay regularization, wherein each network output fits a different "task" (i.e., a different set of labels on the same data points). We show that the solutions to such multi-task training problems can differ dramatically from those obtained by fitting separate neural networks to each task individually. Unlike standard intuitions Caruana (1997) and existing theory Ben-David and Schuller (2003); Maurer et al. (2016) regarding the effects and benefits of multi-task learning, our results do not rely on similarity between tasks.

We focus on shallow, vector-valued (multi-output) neural networks with Rectified Linear Unit (ReLU) activation functions, which are functions $f_{\boldsymbol{\theta}} : \mathbb{R}^d \to \mathbb{R}^T$ of the form

$$f_{\boldsymbol{\theta}}(\boldsymbol{x}) = \sum_{k=1}^{K} \boldsymbol{v}_k \big( \boldsymbol{w}_k^\top \boldsymbol{x} + b_k \big)_+ + \boldsymbol{A}\boldsymbol{x} + \boldsymbol{c} \tag{1}$$

38th Conference on Neural Information Processing Systems (NeurIPS 2024).

where $(\cdot)_+ = \max\{0, \cdot\}$ is the ReLU activation function and $\boldsymbol{w}_k \in \mathbb{R}^d$, $\boldsymbol{v}_k \in \mathbb{R}^T$, and $b_k \in \mathbb{R}$ are the input and output weights and bias of the $k^{\text{th}}$ neuron. $K$ is the number of neurons and $T$ denotes the number of tasks (outputs) of the neural network. The affine term $\boldsymbol{A}\boldsymbol{x} + \boldsymbol{c}$ is the residual connection (or skip connection), where $\boldsymbol{A} \in \mathbb{R}^{T \times d}$ and $\boldsymbol{c} \in \mathbb{R}^T$. The set of all parameters is denoted by $\boldsymbol{\theta} := \left(\{\boldsymbol{v}_k, \boldsymbol{w}_k, b_k\}_{k=1}^K, \boldsymbol{A}, \boldsymbol{c}\right)$.

Neural networks are trained to fit data using gradient descent methods and often include a form of regularization called *weight decay*, which penalizes the $\ell^2$ norm of the network weights. We consider weight decay applied only to the input and output weights of the neurons—no regularization is applied to the biases or residual connection. This is a common setting studied frequently in past work Savarese et al. (2019); Ongie et al. (2019); Parhi and Nowak (2021). Intuitively, only the input and output weights—not the biases or residual connection—affect the "regularity" of the neural network function as measured by its second (distributional) derivative, which is why it makes sense to regularize only these parameters. Given a set of training data points $(\boldsymbol{x}_1, \boldsymbol{y}_1), \ldots, (\boldsymbol{x}_N, \boldsymbol{y}_N) \in \mathbb{R}^d \times \mathbb{R}^T$ and a fixed width[1] $K \geq N^2$, we consider the weight decay interpolation problem:

$$\min_{\boldsymbol{\theta}} \sum_{k=1}^K \|\boldsymbol{v}_k\|_2^2 + \|\boldsymbol{w}_k\|_2^2 \text{ , subject to } f_{\boldsymbol{\theta}}(\boldsymbol{x}_i) = \boldsymbol{y}_i, \ i = 1, \ldots, N \ . \tag{2}$$

By homogeneity of the ReLU activation function (meaning that $(\alpha x)_+ = \alpha(x)_+$ for any $\alpha \geq 0$), the input and output weights of any ReLU neural network can be rescaled as $\boldsymbol{w}_k \mapsto \boldsymbol{w}_k / \|\boldsymbol{w}_k\|_2$ and $\boldsymbol{v}_k \mapsto \boldsymbol{v}_k \|\boldsymbol{w}_k\|_2$ without changing the function that the network represents. Using this fact, several previous works Grandvalet (1998); Grandvalet and Canu (1998); Neyshabur et al. (2015); Parhi and Nowak (2023); Shenouda et al. (2024) note that problem (2) is equivalent to

$$\min_{\boldsymbol{\theta}} \sum_{k=1}^K \|\boldsymbol{v}_k\|_2, \text{ subject to } \{\|\boldsymbol{w}_k\|_2 = 1\}_{k=1}^K, \ f_{\boldsymbol{\theta}}(\boldsymbol{x}_i) = \boldsymbol{y}_i, \ i = 1, \ldots, N \tag{3}$$

in that the minimal objective values of both training problems are the same, and any network $f_{\boldsymbol{\theta}}$ which solves (2) also solves (3), while any $f_{\boldsymbol{\theta}}$ which solves (3) also solves (2) after rescaling of the input and output weights. The regularizer $\sum_{k=1}^K \|\boldsymbol{v}_k\|_2$ is reminiscent of the multi-task lasso Obozinski et al. (2006). It has recently been shown to promote *neuron sharing* in the network, meaning that only a few neurons contribute to all tasks Shenouda et al. (2024).

The optimizations in (2) and (3) are non-convex and may have multiple global minimizers. As an example, consider the single-task, univariate dataset in Fig. 1. For this dataset, (3) has infinitely many global solutions Savarese et al. (2019); Ergen and Pilanci (2021); Debarre et al. (2022); Hanin (2021). Two of the global minimizers are shown in Fig. 1. In some scenarios, the solution on the right may be preferable to the one on the left, since the interpolation function stays closer to the training data points, and hence is more adversarially robust by most definitions Carlini et al. (2019). Moreover, recent theoretical work shows that this solution has other favorable generalization and robustness properties Joshi et al. (2024). Current training methods, however, might produce any one of the infinite number of solutions, depending on the random initialization of the parameters as well as other possible sources of randomness in the training process. It is impossible to control this using existing training algorithms, which might explain many problems associated with current neural networks such as their sensitivity to adversarial attacks. In contrast, as we show in this paper, training a network to interpolate the data in Fig. 1 along with additional interpolation tasks with different labels almost always produces a unique solution, given by the (potentially preferable) interpolation depicted on the right. This demonstrates that the solutions to multi-task learning problems can be profoundly different than those of single-task learning problems.

The main contributions of our paper are:

**Uniqueness of Multi-task Solutions.** In the univariate setting ($d = 1$) we prove that the solutions to multi-task learning problems with different tasks almost always represent a unique function, and we give a precise condition for the exceptional cases where solutions are non-unique.

---

[1]By an argument similar to the proof of Theorem 5 of Shenouda et al. (2024), as long as $K \geq N^2$, problem (3) is well-posed and attains the same minimal objective value (regardless of which $K \geq N^2$ is chosen). Therefore, in this work, we always assume that $K$ is some fixed value larger than $N^2$.

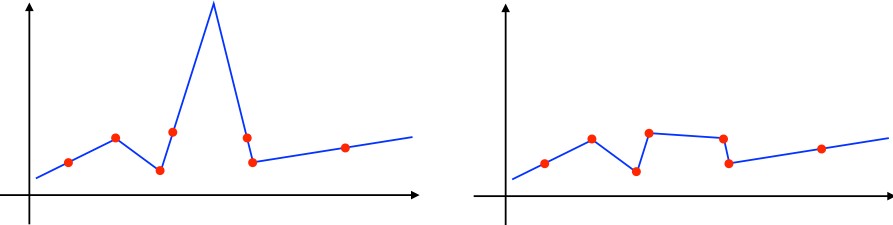

Figure 1: Two solutions to ReLU neural network interpolation (blue) of training data (red). The functions on the left and right both interpolate the data and both are global minimizers of (2) and (3), and minimize the second-order total variation of the interpolation function Parhi and Nowak (2021). In fact, all convex combinations of the two solutions above are also global solutions to both training problems.

**Multi-task Training ≡ Kernel Method (almost always).** When the solution to the univariate weight decay problem is unique, it is given by the connect-the-dots interpolant of the training data points: i.e., the optimal solution is a linear spline which performs straight-line interpolation between consecutive data points in all tasks. On the support of the data, this solution agrees with the minimum-norm interpolant in the first-order Sobolev space $H^1$, a reproducing kernel Hilbert space (RKHS) which contains all functions with first derivatives in $L^2$ De Boor and Lynch (1966). In contrast, solutions to the single-task learning problem are non-unique in general and are given by minimum-norm interpolating functions in the non-Hilbertian Banach space $\mathrm{BV}^2$ Parhi and Nowak (2021), which contains all functions with second distributional derivatives[2] in $L^1$. This shows that the individual task solutions to a multi-task learning problem are almost always equivalent to those of a minimum-norm kernel interpolation problem, whereas single-task solutions generally are not.

**Insights on Multivariate Multi-Task Problems.** We provide empirical evidence and mathematical analysis which indicate that similar conclusions hold in multivariate settings. Specifically, the individual task solutions to a multi-task learning problem are approximately minimum-norm solutions in a particular RKHS determined by the optimal neurons. In contrast, learning each task in isolation results in solutions that are minimum-norm with respect to a non-Hilbertian Banach norm over the optimal neurons.

## 2 Related Works

**Characterizations of ReLU neural network solutions:** Hanin (2021); Stewart et al. (2023) characterized the neural network solutions to (2) in the univariate input/output setting. Boursier and Flammarion (2023) showed that in the univariate input/output case, when weight decay is modified to include the biases of each neuron, the solution is unique. Moreover, under certain assumptions, it is the sparsest interpolant (i.e., the interpolant with the fewest neurons). Our work differs from these in that we study the multi-task setting, showing that univariate-input multi-task solutions are almost always unique and equivalent to the connect-the-dots solution, which is generally *not* the sparsest, and is a minimum-norm solution in a Sobolev RKHS. While characterizing solutions to (2) in the multivariate setting is more challenging, there exist some results under certain dataset assumptions Ardeshir et al. (2023); Zeno et al. (2024) or by leveraging a convex reformulation Ergen and Pilanci (2021); Mishkin and Pilanci (2023) of (2).

**Function spaces associated with neural networks:** For single-output ReLU neural networks, Savarese et al. (2019); Ongie et al. (2019) related weight decay regularization on the parameters of the model to regularizing a particular semi-norm on the neural network function. Ongie et al. (2019) showed that this semi-norm is not an RKHS semi-norm, highlighting a fundamental difference between learning with neural networks and kernel methods. Parhi and Nowak (2021, 2022); Bartolucci et al. (2023); Unser (2021) studied the function spaces associated with this semi-norm, and developed representer theorems showing that optimal solutions to the minimum-norm data fitting problem over

---

[2]Technically, $\mathrm{BV}^2$ contains all functions with second distributional derivatives in $\mathcal{M}$, the space of Radon measures with finite total variation. $\mathcal{M}$ can be viewed as a "generalization" of $L^1$ (see Ch. 7.3, p.223 in Folland (1999)).

these spaces are realized by finite-width ReLU networks. Consequently, finite-width ReLU networks trained with weight decay are optimal solutions to the regularized data-fitting problem posed over these spaces. Function spaces and representer theorems for multi-output and deep neural networks were later developed in Korolev (2022); Parhi and Nowak (2022); Shenouda et al. (2024).

**Multi-Task Learning:**     The advantages of multi-task learning have been extensively studied in the machine learning literature Obozinski et al. (2006, 2010); Argyriou et al. (2006, 2008); Caruana (1997). In particular, the theoretical properties of multi-task neural networks have been studied in Lindsey and Lippl (2023); Collins et al. (2024); Shenouda et al. (2024). The underlying intuition in these past works has been that learning multiple related tasks simultaneously can help select or learn the most useful features for all tasks. Our work differs from this traditional paradigm as we consider multi-task neural networks trained on very general tasks which may be diverse and unrelated.

## 3    Univariate Multi-Task Neural Network Solutions

For any function $f$ that can be represented by a neural network (1) with width $K$, we define its representational cost to be

$$R(f) := \inf_{\boldsymbol{\theta}} \sum_{k=1}^{K} \|\boldsymbol{v}_k\|_2 \text{ , subject to } \|\boldsymbol{w}_k\|_2 = 1 \, \forall k, \, f = f_{\boldsymbol{\theta}} \tag{4}$$

where $\boldsymbol{\theta} = \left(\{\boldsymbol{v}_k, \boldsymbol{w}_k, b_k\}_{k=1}^{K}, \boldsymbol{A}, \boldsymbol{c}\right)$. Taking an inf over all possible neural network parameters is necessary as there are multiple neural networks which can represent the same function. Solutions to (3) minimize this representational cost subject to the data interpolation constraint. This section gives a precise characterization of the solutions to the multi-task neural network interpolation problem in the univariate setting ($d = 1$).

For the training data points $(x_1, \boldsymbol{y}_1), \ldots, (x_N, \boldsymbol{y}_N) \in \mathbb{R} \times \mathbb{R}^T$, let $y_{it}$ denote the $t^{\text{th}}$ coordinate of the label vector $\boldsymbol{y}_i$. For each $t = 1, \ldots, T$, let $\mathcal{D}_t$ denote the univariate dataset $(x_1, y_{it}), \ldots, (x_N, y_{Nt}) \in \mathbb{R} \times \mathbb{R}$, and let

$$s_{it} = \frac{y_{i+1,t} - y_{it}}{x_{i+1} - x_i} \tag{5}$$

denote the slope of the straight line between $(x_i, y_{it})$ and $(x_{i+1}, y_{i+1t})$. The connect-the-dots interpolant of the dataset $\mathcal{D}_t$ is the function $f_{\mathcal{D}_t}$ which connects the consecutive points in dataset $\mathcal{D}_t$ with straight lines (see Fig. 2). Its slopes on $(-\infty, x_2]$ and $[x_{N-1}, \infty)$ are $s_{1t}$ and $s_{N-1t}$, respectively. In the following section, we state a simple necessary and sufficient condition under which the connect-the-dots interpolation $f_{\mathcal{D}} = (f_{\mathcal{D}_1}, \ldots, f_{\mathcal{D}_T})$ is the *unique* optimal interpolant of the datasets $\mathcal{D}_1, \ldots, \mathcal{D}_T$. We also demonstrate that the set of multi-task datasets which satisfy the necessary condition for non-uniqueness, viewed as a subset of $\mathbb{R}^N \times \mathbb{R}^{T \times N}$, has Lebesgue measure zero.

This result raises an interesting new connection between data fitting with ReLU neural networks and traditional kernel-based learning methods. Indeed, connect-the-dots interpolation is also the minimum-norm interpolant over the first-order Sobolev space $H^1([x_1, x_N])$, itself an RKHS whose norm penalizes the $L^2$ norm of the derivative of the function. In particular, $f_{\mathcal{D}_t}$ agrees on $[x_1, x_N]$ with the function $f(x) = \sum_{j=1}^{N} \alpha_j k(x, x_j)$ whose coefficients $\alpha_j$ solve the kernel optimization problem

$$\min_{\alpha_1, \ldots, \alpha_N \in \mathbb{R}} \sum_{i=1}^{N} \sum_{j=1}^{N} \alpha_i \alpha_j k(x_i, x_j) \text{ , subject to } \sum_{j=1}^{N} \alpha_j k(x_i, x_j) = y_{it}, \, i = 1, \ldots, N \, . \tag{6}$$

with the kernel $k(x, x') = 1 - (x - x')_+ + (x - x_1)_+ + (x_1 - x')_+$ De Boor and Lynch (1966). Therefore, our result shows that the individual outputs of solutions to (3) for $T > 1$ tasks almost always coincide on $[x_1, x_N]$ with this kernel solution; for example, this occurs with probability one if the task labels are sampled from an absolutely continuous distribution. In contrast, optimal solutions to the (3) in the case $T = 1$ are generally non-unique and may not coincide with the connect-the-dots kernel solution Hanin (2022). We note that for $T = 1$, our result is consistent with the characterization of univariate solutions to (3) in Hanin (2022).

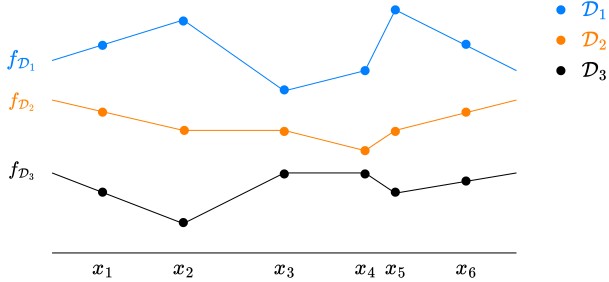

Figure 2: The connect-the-dots interpolant $f_\mathcal{D} = (f_{\mathcal{D}_1}, f_{\mathcal{D}_2}, f_{\mathcal{D}_3})$ of three datasets $\mathcal{D}_1, \mathcal{D}_2, \mathcal{D}_3$.

## 3.1 Characterization and Uniqueness

Our main result is stated in the following theorem:

**Theorem 3.1.** *The connect-the-dots function $f_\mathcal{D}$ is always a solution to (3). Moreover, the solution to problem (3) is non-unique if and only if the following condition is satisfied: for some $i = 2, \ldots, N-2$, the two vectors*

$$s_i - s_{i-1} = \frac{y_{i+1} - y_i}{x_{i+1} - x_i} - \frac{y_i - y_{i-1}}{x_i - x_{i-1}} \tag{7}$$

*and*

$$s_{i+1} - s_i = \frac{y_{i+2} - y_{i+1}}{x_{i+2} - x_{i+1}} - \frac{y_{i+1} - y_i}{x_{i+1} - x_i} \tag{8}$$

*are both nonzero and aligned.*[3] *If this condition is not satisfied, then $f_\mathcal{D}$ is the unique solution to (3). Furthermore, as long as $T > 1$ and $N > 1$, the set of all possible data points $x_1, \ldots, x_N \in \mathbb{R}$ and $y_1, \ldots, y_N \in \mathbb{R}^T$ which admit non-unique solutions has Lebesgue measure zero (as a subset of $\mathbb{R}^N \times \mathbb{R}^{T \times N}$).*

**Corollary 1.** *If $T > 1$ and $N > 1$ and the data points $x_1, \ldots, x_N \in \mathbb{R}$ and label vectors $y_1, \ldots, y_N \in \mathbb{R}^T$ are sampled from an absolutely continuous distribution with respect to the Lebesgue measure on $\mathbb{R}^N \times \mathbb{R}^{T \times N}$, then with probability one, the connect-the-dots function $f_\mathcal{D}$ is the unique solution to (3).*

**Remark 1.** *The proof of Theorem 3.1, which relies mainly on Theorem 3.2 as we describe below, also characterizes solutions of the regularized loss problem*

$$\min_{\boldsymbol{\theta}} \sum_{i=1}^{N} \mathcal{L}(f_{\boldsymbol{\theta}}(x_i), y_i) + \lambda \sum_{k=1}^{K} \|v_k\|_2 \quad \text{subject to} \quad |w_k| = 1, \ k = 1, \ldots, K \tag{9}$$

*for input dimension $d = 1$, any $\lambda > 0$, and any loss function $\mathcal{L}$ which is lower semicontinuous in its second argument. Specifically, any $f_{\boldsymbol{\theta}}$ which solves (9) is linear between consecutive data points $[x_i, x_{i+1}]$ unless the vectors $\hat{s}_i - \hat{s}_{i-1}$ and $\hat{s}_{i+1} - \hat{s}_i$ are both nonzero and aligned, where $\hat{s}_i := \frac{\hat{y}_{i+1} - \hat{y}_i}{x_{i+1} - x_i}$ and $\hat{y}_i := f_{\boldsymbol{\theta}}(x_i)$.*

Previous works Shenouda et al. (2024) and Lindsey and Lippl (2023) showed that multi-task learning encourages *neuron sharing*, where all task are encouraged to utilize the same set of neurons or representations. Our result above shows that univariate multi-task training is an extreme example of this phenomenon, since $f_\mathcal{D}$ can be represented using only $N - 1$ neurons, all of which contribute to all of the network outputs. Therefore, in the scenario we study here, neuron sharing almost always occurs even if the tasks are unrelated.

The full proof of Theorem 3.1 appears in Appendix A.1. We outline the main ideas here. Our proof relies on the fact that any $\mathbb{R} \to \mathbb{R}^T$ ReLU neural network of the form (1) which solves (3) represents $T$ continuous piecewise linear (CPWL) functions, where the change in slope of the $t^\text{th}$ function at the $k^\text{th}$ knot is equivalent the $t^\text{th}$ entry of the $k^\text{th}$ output weight vector (see Appendix A.1 for further detail). This fact allows us to draw a one-to-one correspondence between each knot in the function and each neuron in the neural network. The proof relies primarily on the following lemma:

---

[3]Two vectors $u_1$ and $u_2$ are *aligned* if $u_1^\top u_2 = \|u_1\|\|u_2\|$.

**Lemma 3.2.** *Let $f : \mathbb{R} \to \mathbb{R}^T$ be a function for which each output $f_t$ is CPWL and interpolates $\mathcal{D}_t$. Suppose that at some $\tilde{x}$ between consecutive data points, one or more of the outputs $f_t$ has a knot. Let $\tilde{x}_1$ and $\tilde{x}_2$ be the $x$-coordinates of the closest knots before and after $\tilde{x}$, respectively. Denote the slopes of $f_t$ around this interval in terms of $a_t$, $b_t$, $c_t$, and $\delta_t$ as in Fig. 3, and let $\boldsymbol{a}, \boldsymbol{b}, \boldsymbol{c}, \boldsymbol{\delta} \in \mathbb{R}^T$ be the vectors containing the respective values for each task.*

*Then removing the knot at $x$ and instead connecting $x_i$ and $x_{i+1}$ by a straight line does not increase $R(f)$. Furthermore, if $\boldsymbol{a} - \boldsymbol{b}$ and $\boldsymbol{b} - \boldsymbol{c}$ are not aligned, then doing so will strictly decrease $R(f)$.*

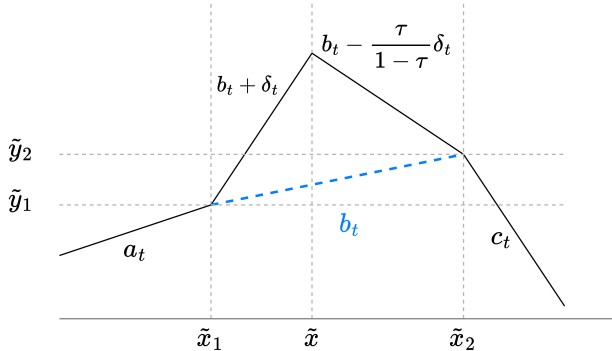

Figure 3: The function output $f_t$ around the knot at $\tilde{x}$, where $\tau = \frac{\tilde{x} - \tilde{x}_1}{\tilde{x}_2 - \tilde{x}_1}$. Each line segment in the figure is labeled with its slope. For any particular output $t$, it may be the case that $f_t$ does not have a knot at $\tilde{x}$ (in which case $\delta_t = 0$); does not have a knot at $\tilde{x}_1$ (in which case $a_t = b_t + \delta_t$); and/or does not have a knot at $\tilde{x}_2$ (in which case $b_t - \frac{\tau}{1-\tau}\delta_t = c_t$).

*Proof of Theorem 3.2.* The contribution of these knots to $R(f)$ is:

$$
\begin{aligned}
& \|\boldsymbol{\delta} + \boldsymbol{b} - \boldsymbol{a}\|_2 \;+\; \frac{1}{1-\tau}\|\boldsymbol{\delta}\|_2 \;+\; \|\boldsymbol{c} - \boldsymbol{b} + \tau\boldsymbol{\delta}/(1-\tau)\|_2 \\
\geq\; & \|\boldsymbol{b} - \boldsymbol{a}\|_2 - \|\boldsymbol{\delta}\|_2 \;+\; \frac{1}{1-\tau}\|\boldsymbol{\delta}\|_2 \;+\; \|\boldsymbol{c} - \boldsymbol{b}\|_2 - \frac{\tau}{1-\tau}\|\boldsymbol{\delta}\|_2 \qquad (10)\\
=\; & \|\boldsymbol{b} - \boldsymbol{a}\|_2 \;+\; \|\boldsymbol{c} - \boldsymbol{b}\|_2
\end{aligned}
$$

where the inequality follows from the triangle inequality. This shows that taking $\delta_t = 0$ for all outputs, which corresponds to connecting $\tilde{x}_1$ and $\tilde{x}_2$ with a straight line in all outputs, will never increase the representational cost of $f$. The triangle inequality used in (10) holds with equality for some $\boldsymbol{\delta} \neq \boldsymbol{0}$ if and only if $\boldsymbol{a} - \boldsymbol{b}$, $\boldsymbol{b} - \boldsymbol{c}$, and $\boldsymbol{\delta}$ are aligned with $\|\boldsymbol{\delta}\|_2 \leq \min\{\|\boldsymbol{a} - \boldsymbol{b}\|_2, \frac{1-\tau}{\tau}\|\boldsymbol{b} - \boldsymbol{c}\|_2\}$. $\qquad\square$

Theorem 3.2 states that removing neurons which are located away from the data points and replacing them with a straight line will never increase the cost of the network, and it will strictly decrease the cost unless $\boldsymbol{a} - \boldsymbol{b}$ and $\boldsymbol{b} - \boldsymbol{c}$ are aligned. This result implies that the connect-the-dots interpolant $f_{\mathcal{D}}$ is always a solution to (3), since we may take any solution $f$ of (3) and remove all knots from it (resulting in the function $f_{\mathcal{D}}$) without increasing its representational cost. If $\boldsymbol{s}_i - \boldsymbol{s}_{i-1}$ and $\boldsymbol{s}_{i+1} - \boldsymbol{s}_i$ are aligned for some $i = 2, \ldots, N - 2$, we can view any interpolant on the interval $[x_i, x_{i+1}]$ as an instance of Fig. 3 with $\boldsymbol{a} = \boldsymbol{s}_{i-1}$, $\boldsymbol{b} = \boldsymbol{s}_i$, and $\boldsymbol{c} = \boldsymbol{s}_{i+1}$. By Theorem 3.2, any CPWL function with a knot at some point $\tilde{x} \in (x_i, x_{i+1})$ can have the same representational cost as the connect-the-dots solution on this interval, only if $\boldsymbol{a} - \boldsymbol{b}$ and $\boldsymbol{b} - \boldsymbol{c}$ are aligned.

We can also prove by contradiction that optimal solutions are unique on $[x_i, x_{i+1}]$ as long as $\boldsymbol{s}_i - \boldsymbol{s}_{i-1}$ and $\boldsymbol{s}_{i+1} - \boldsymbol{s}_i$ are *not* aligned. Suppose that there is some other optimal interpolant $f$ which is *not* the connect-the-dots solution $f_{\mathcal{D}}$ on an interval $[x_i, x_{i+1}]$ for which $\boldsymbol{s}_i - \boldsymbol{s}_{i-1}$ and $\boldsymbol{s}_{i+1} - \boldsymbol{s}_i$ are not aligned. Then apply the lemma repeatedly to remove all knots from $f_{\boldsymbol{\theta}}$ which are not located at the data points, except for a single remaining knot at some $\tilde{x}$ between consecutive data points. If this knot occurs after $x_2$ or before $x_{N-1}$, the lemma implies automatically (again taking $\boldsymbol{a} = \boldsymbol{s}_{i-1}$, $\boldsymbol{b} = \boldsymbol{s}_i$, and $\boldsymbol{s}_{i+1}$) that removing this knot would strictly decrease the representational cost of the function, contradicting optimality of $f$. To conclude the proof, it remains only to show that any optimal

interpolant of the dataset must agree with the connect-the-dots interpolant $f_{\mathcal{D}}$ before $x_2$ and after $x_{N-1}$; the details of this argument appear in Appendix A.1. As our theorem and corollary quantify, real-world regression datasets (which are typically real-valued and often assumed to incorporate some random noise from an absolutely continuous distribution, e.g. Gaussian) are extremely unlikely to satisfy this special alignment condition; hence, our claim that connect-the-dots interpolation is almost always the unique solution to (3).

## 4 Multivariate Multi-Task Neural Network Training

In Section 3, we proved that univariate-input functions learned by neural networks trained on multiple tasks simultaneously can be profoundly different from the functions learned by networks trained on each task separately. In this section, we demonstrate an extension of this phenomenon to the multivariate-input case. Similar to the univariate case, the multivariate single-task weight decay regularized learning problem corresponds to a norm-penalized learning problem in a non-Hilbertian Banach space, where solutions may be non-unique Parhi and Nowak (2021). Indeed, Figure 4 illustrates a multivariate single-task dataset with multiple min-norm interpolants. In this section, we show that, as in the univariate case, multivariate multi-task learning can produce solutions that are strikingly different from the corresponding single-task learning solutions. Moreover, we show that the multivariate multi-task learning problem can also be related to a norm-penalized learning problem over an RKHS. Here we analyze neural networks of the form

$$f_{\boldsymbol{\theta}}(\boldsymbol{x}) = \sum_{k=1}^{K} \boldsymbol{v}_k \big(\boldsymbol{w}_k^\top \boldsymbol{x} + b_k\big)_+ \tag{11}$$

where $\boldsymbol{w}_k \in \mathbb{S}^{d-1}$, $b_k \in \mathbb{R}$, $\boldsymbol{v}_k \in \mathbb{R}^T$, and $\boldsymbol{\theta} := \{\boldsymbol{v}_k, \boldsymbol{w}_k, b_k\}_{k=1}^{K}$. Since the analysis in this section is not dependent on the residual connection, we omit it for ease of exposition. We consider the multivariate-input, $T$-task neural network learning problem

$$\min_{\boldsymbol{\theta}} \sum_{i=1}^{N} \mathcal{L}\left(\boldsymbol{y}_i, f_{\boldsymbol{\theta}}(\boldsymbol{x}_i)\right) + \lambda \sum_{k=1}^{K} \|\boldsymbol{v}_k\|_2 \tag{12}$$

for some dataset $(\boldsymbol{x}_1, \boldsymbol{y}_1), \ldots, (\boldsymbol{x}_N, \boldsymbol{y}_N) \in \mathbb{R}^d \times \mathbb{R}^T$, where $\mathcal{L}$ is any loss function which is lower semicontinuous in its second argument and separable across the $T$ tasks, and $K \geq N^2$ (see [1]).

To characterize the nature of the functions whose parameters solve (12), note that the optimal task $s$ output weights $v_{1s}^*, \ldots, v_{Ks}^*$ for (12) also minimize

$$J(v_{1s}, \ldots, v_{Ks}) := \sum_{i=1}^{N} \mathcal{L}\left(y_{is}, \sum_{k=1}^{K} v_{ks}\boldsymbol{\Phi}_{ik}\right) + \lambda \sum_{k=1}^{K} \left\|\begin{bmatrix} v_{ks} \\ \boldsymbol{v}_{k\backslash s}^* \end{bmatrix}\right\|_2 \tag{13}$$

where $\boldsymbol{v}_{k\backslash s}^*$ denotes the vector $\boldsymbol{v}_k^*$ with its $s^{\text{th}}$ element $v_{ks}^*$ excluded and $\boldsymbol{\Phi} \in \mathbb{R}^{N \times K}$ is a matrix whose $i, k^{\text{th}}$ entry is $\boldsymbol{\Phi}_{ik} = \big(\boldsymbol{x}_i^\top \boldsymbol{w}_k^* + b_k^*\big)_+$. Thus, $J$ is the objective function of (12) with all parameters except for $v_{1s}, \ldots, v_{Ks}$ held fixed at their optimal values. Note that if $v_{1s}^*, \ldots, v_{Ks}^*$ did not minimize $J$, they would not be optimal for (12).

We are interested in analyzing the behavior of solutions to (12) as the number of tasks $T$ grows. Intuitively, if $T$ is very large, it it is reasonable to expect that the optimal output weight $v_{ks}^*$ for an individual neuron $k$ and task $s$ would be relatively small compared to the sum of the output weights $v_{kt}^*$ for tasks $t \neq s$. In this case, the $k^{\text{th}}$ term of the regularizer in (12) would be approximately equal to

$$\|\boldsymbol{v}_k^*\|_2 = \sqrt{(v_{ks}^*)^2 + \|\boldsymbol{v}_{k\backslash s}^*\|_2^2} \approx \|\boldsymbol{v}_{k\backslash s}^*\|_2 + \frac{(v_{ks}^*)^2}{2\|\boldsymbol{v}_{k\backslash s}^*\|_2} \tag{14}$$

for any individual task $s$, where $\|\boldsymbol{v}_{k\backslash s}^*\|_2^2 := \sum_{t\neq s}(v_{ks}^*)^2$. The approximation above comes from the Taylor expansion $f(x) = \sqrt{x^2 + c^2} = c + \frac{x^2}{2c} - \frac{x^4}{8c^3} + \frac{x^6}{16c^5} - \ldots$, whose higher order terms quickly become negligible if $0 < x \ll c$. Notice that the right hand side of (14) is a quadratic function of $v_{ks}^*$, which suggests that the regularization term of (13) resembles a weighted $\ell^2$ regularizer when $v_{ks}$ is close to its optimal value $v_{ks}^*$.

The above reasoning can be made precise by observing that, in multi-task learning problems, the order in which the tasks are assigned to the network outputs is irrelevant. Therefore, we can assume the tasks are assigned uniformly at random to each of the networks outputs. This random assignment process induces a distribution on the training data (labels) for each output $\boldsymbol{y}_{\cdot,1}, \ldots, \boldsymbol{y}_{\cdot,T}$. Under this assumption, the following are true with high probability as $T$ grows (see Nakhleh et al. (2024), Appendix A.2 for further detail):

1. The magnitude of $v_{ks}^*$ is dominated by $\|\boldsymbol{v}_{k \backslash s}^*\|_2$, which implies that the remainder in the quadratic Taylor series approximation tends to zero.

2. $\|\boldsymbol{v}_{k \backslash s}^*\|_2$ concentrates around the norm of the full vector of output weights $\|\boldsymbol{v}_k^*\|_2$, which means that the Taylor approximation tends to the same quadratic function for all tasks.

A more detailed analysis is given in the companion paper Nakhleh et al. (2024); here we informally state a theorem summarizing the main conclusion.

**Theorem 4.1.** *Nakhleh et al. (2024) For an individual task $s$, consider the objective*[4]

$$H(v_{1s}, \ldots, v_{Ks}) := \sum_{i=1}^{N} \mathcal{L}\left(y_{is}, \sum_{k \in \mathcal{S}} v_{ks} \boldsymbol{\Phi}_{ik}\right) + \lambda \sum_{k \in \mathcal{S}} \left(\|\boldsymbol{v}_{k \backslash s}^*\|_2 + \frac{v_{ks}^2}{2\|\boldsymbol{v}_{k \backslash s}^*\|_2}\right) \quad (15)$$

*Then as $T$ grows, the global minimizer $v'_{1s}, \ldots, v'_{Ks}$ of $H$ satisfies*

$$|J(v'_{1s}, \ldots, v'_{Ks}) - J(v_{1s}^*, \ldots, v_{Ks}^*)| \rightarrow 0 \quad (16)$$

*with probability approaching one.*

The theorem states that the solution to the followed weighted $\ell^2$ regularized problem

$$\min_{v_{1s}, \ldots, v_{Ks}} \sum_{i=1}^{N} \mathcal{L}\left(y_{is}, \sum_{k \in \mathcal{S}} v_{ks} \boldsymbol{\Phi}_{ik}\right) + \frac{\lambda}{2} \sum_{k \in \mathcal{S}} \gamma_{ks} v_{ks}^2 \quad (17)$$

where $\gamma_{ks} := 1/\|\boldsymbol{v}_{k/s}^*\|_2$ is an approximate minimizer of (13), with stronger approximation as $T$ increases. In contrast, when $T = 1$, the optimization

$$\min_{v_1, \ldots, v_K} \sum_{i=1}^{N} \mathcal{L}\left(y_i, \sum_{k=1}^{K} v_k \boldsymbol{\Psi}_{ik}\right) + \lambda \sum_{k=1}^{K} |v_k| \quad (18)$$

yields output weights which are exactly optimal for (12). Note that the matrices $\boldsymbol{\Phi}$ in (17) and $\boldsymbol{\Psi}$ in (18) are not the same, since they are determined by the optimal input weights and biases for (12), which are themselves data- and task-dependent. Nonetheless, comparing (17) and (18) highlights the different nature of solutions learned for (12) in the single-task versus multi-task case. The multi-task learning problem favors linear combinations of the optimal neurons which have a minimal weighted $\ell^2$ regularization penalty. In contrast, the single-task learning problem favors linear combinations of optimal neurons which have a minimal $\ell^1$ penalty. Therefore, multi-task learning with a large number of tasks promotes a fundamentally different linear combination of the optimal features learned in the hidden layer.

To gain further insight, note that concentration of $\|\boldsymbol{v}_{k \backslash s}^*\|_2$ around $\|\boldsymbol{v}_k^*\|_2$ implies that for large $T$:

$$\gamma_{ks} \approx \gamma_k := \frac{1}{\|\boldsymbol{v}_k^*\|_2}. \quad (19)$$

(see Lemma 4.2 in Nakhleh et al. (2024)). This reveals a novel connection between the problem of minimizing (13) and a norm-regularized data fitting problem in an RKHS. Specifically, consider the finite-dimensional linear space

$$\mathcal{H} := \left\{f_{\boldsymbol{v}} = \sum_{k=1}^{K} v_k \phi_k : \boldsymbol{v} \in \mathbb{R}^K\right\} \quad (20)$$

---

[4]$\mathcal{S}$ denotes the set of optimal active neurons for (12). A neuron $\eta(\boldsymbol{x}) = \boldsymbol{v}(\boldsymbol{w}^T \boldsymbol{x} + b)_+$ is *active* if $\|\boldsymbol{v}\|_2 > 0$.

where $\phi_k(\boldsymbol{x}) = (\boldsymbol{w}_k^* \cdot \boldsymbol{x} + b_k^*)_+$, equipped with the inner product

$$\langle f_{\boldsymbol{v}}, f_{\boldsymbol{u}} \rangle_{\mathcal{H}} = \boldsymbol{v}^\top \boldsymbol{Q} \boldsymbol{u} \tag{21}$$

where $\boldsymbol{Q} = \mathrm{diag}\left(\frac{\gamma_1}{2}, \ldots, \frac{\gamma_K}{2}\right)$. As a finite-dimensional inner product space, $\mathcal{H}$ is necessarily a Hilbert space; furthermore, finite-dimensionality of $\mathcal{H}$ implies that all linear functionals (including the point evaluation functional) on $\mathcal{H}$ are continuous. Therefore, $\mathcal{H}$ is an RKHS, with reproducing kernel

$$\kappa(\boldsymbol{x}, \boldsymbol{x}') = \sum_{k=1}^{K} \phi_k(\boldsymbol{x}) Q_{kk}^{-1} \phi_k(\boldsymbol{x}'). \tag{22}$$

Note that $\kappa$ indeed satisfies the reproducing property, that is, $\langle \kappa(\cdot, \boldsymbol{x}), f \rangle_{\mathcal{H}} = f(\boldsymbol{x})$ for any $f \in \mathcal{H}$ and any $\boldsymbol{x}$. To see this, write

$$\langle \kappa(\cdot, \boldsymbol{x}), f \rangle_{\mathcal{H}} = \left\langle \sum_{k=1}^{K} Q_{kk}^{-1} \phi_k(\boldsymbol{x}) \phi_k, \sum_{k=1}^{K} v_k \phi_k \right\rangle_{\mathcal{H}}. \tag{23}$$

We can view the term on the left as a function $g_{\boldsymbol{u}} \in \mathcal{H}$ where $u_k = Q_{kk}^{-1} \phi_k(\boldsymbol{x})$ or $\boldsymbol{u} = \phi(\boldsymbol{x}) \boldsymbol{Q}^{-1}$, so this is equivalent to

$$\langle \kappa(\cdot, \boldsymbol{x}), f \rangle_{\mathcal{H}} = \langle g_{\boldsymbol{u}}, f \rangle_{\mathcal{H}} = \phi(\boldsymbol{x}) \boldsymbol{Q}^{-1} \boldsymbol{Q} \boldsymbol{v} = \sum_{k=1}^{K} v_k \phi(\boldsymbol{x}) = f(\boldsymbol{x}). \tag{24}$$

Finding a minimizer of $H$ over $\mathbb{R}^K$ is thus equivalent to solving

$$\underset{f \in \mathcal{H}}{\arg\min} \sum_{i=1}^{N} \mathcal{L}(y_{is}, f(\boldsymbol{x}_i)) + \lambda \|f\|_{\mathcal{H}}^2. \tag{25}$$

We provide empirical evidence for the claims presented in this section in Figure 4 on a simple multi-variate dataset. First, we demonstrate the variety of min-norm interpolates to this dataset in a single task setting. In contrast, we show that the solutions obtained via multi-task learning with additional random tasks have less variability across different trials and are often much smoother than those obtained by single-task learning supporting our claim that these solutions are well approximated by solving a kernel ridge regression problem. We also verify that the optimization (15) is a good approximation for (13).

## 5 Conclusion and Discussion

We have shown that univariate, multi-task shallow ReLU neural networks which interpolate a dataset with minimal sum of squared weights almost always represent a unique function. This function performs straight-line interpolation between consecutive data points for each task. This solution is also the solution to a min-norm data-fitting problem in an RKHS. We provide mathematical analysis and numerical evidence suggesting that a similar conclusion may hold in the mulvariate-input case, as long as the tasks are sufficiently large in number. These results indicate that multi-task training of neural networks can produce solutions that are strikingly different from those obtained by single-task training, and highlights a novel connection between these multi-task solutions and kernel methods.

Future work could aim to extend these results to deep neural network architectures. We also focus here on characterizing global solutions to the optimizations in (2) and (3). Whether or not networks trained with gradient descent-based algorithms will converge to global solutions remains an open question: our low-dimensional numerical experiments in Sections 3 and 4 indicate that they do, but a more rigorous analysis of the training dynamics would be an interesting separate line of research. Finally, while our analysis and experiments in Section 4 indicate that multi-variate, multi-task neural network solutions behave similarly to $\ell^2$ regression over a fixed kernel, we have not precisely characterized what that kernel is in the multi-input case as we have in the single-input case: developing such a characterization is of interest for future work.

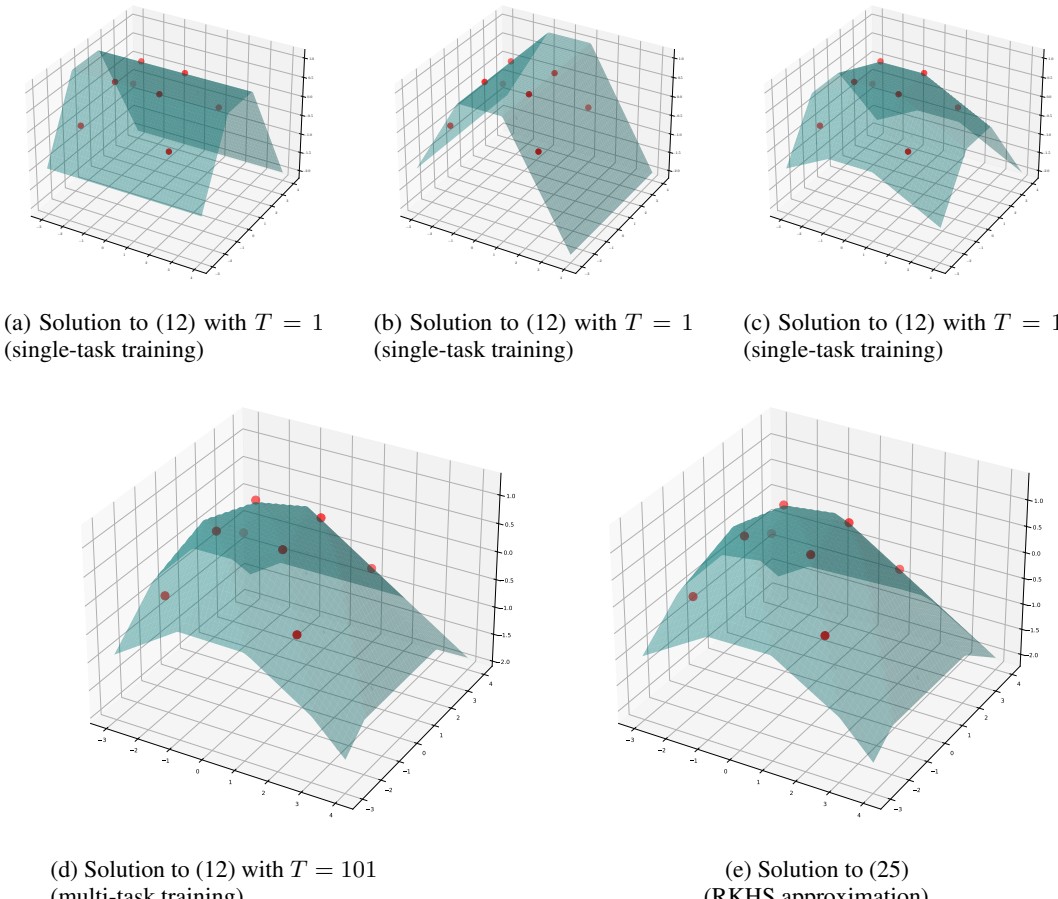

(a) Solution to (12) with $T = 1$ (single-task training)

(b) Solution to (12) with $T = 1$ (single-task training)

(c) Solution to (12) with $T = 1$ (single-task training)

(d) Solution to (12) with $T = 101$ (multi-task training)

(e) Solution to (25) (RKHS approximation)

Figure 4: ReLU network interpolation in two-dimensions. The solutions shown were obtained with regularization parameter $\lambda \approx 0$. *Top Row – Solutions to single-task training*: Figures 4a, 4b and 4c show solutions to ReLU neural network interpolation (blue surface) of training data (red). The eight data points are located at the vertices of two squares, both centered at the origin. The outer square has side-length two and values of $0$ at the vertices. The inner square has side-length one and values of $1$ at the vertices. All three functions interpolate the data and are global minimizers of (2) and (3) when solving for just this task (i.e., $T = 1$). Due to the simplicity of this dataset the optimality of the solutions in the first row were confirmed by solving the equivalent convex optimization to (2) developed in Ergen and Pilanci (2021). *Bottom Row – Solutions to multi-task training:* Figure 4d shows the solution to the first output of a multi-task neural network with $T = 101$ tasks. The first output is the original task depicted in the first row while the labels for other 100 tasks are randomly generated i.i.d from a Bernoulli distribution with equal probability for one and zero. Here we show one representative example; more examples are depicted in Appendix B showing that this phenomenon holds across many runs. Figure 4e shows the solution to fitting the training data by solving (25) over a fixed set of features learned by the multi-task neural network with $T = 100$ random tasks. We observe that unlike the highly variable solutions of single-task optimization problem, the solutions obtained by solving the multi-task optimizations are nearly identical, as one would have for kernel methods. Moreover, the solution obtained by solving (25) is also similar to the solution of the full multi-task training problem with all $T = 101$ tasks.

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

# A Proofs of Main Results

## A.1 Proof of Theorem 3.1

*Proof.* We break the proof into the following sections.

**Unregularized Residual Connection.** We first discuss the utility of the unregularized residual connection in our analysis. Consider a single-input/output function $f$ represented by a ReLU network $f_{\boldsymbol{\theta}} : \mathbb{R} \to \mathbb{R}$ with unit norm input weights. Suppose $f_{\boldsymbol{\theta}}$ includes two neurons,

$$\eta_1(x) = v_1(w_1 x + b_1)_+ \quad \text{and} \quad \eta_2(x) = v_2(w_2 x + b_2)_+$$

with $b_1/w_1 = b_2/w_2$ such that they both activate at the same location. We have the following two cases:

1. **Same Sign Weights:** If $\text{sgn}(w_1) = \text{sgn}(w_2)$, the neurons activate in the same direction and can be merged into a single neuron. This would reduce the representational cost without altering the function.

2. **Opposite Sign Weights:** If $w_1 = 1$ and $w_2 = -1$, their sum can be rewritten using $x = (x)_+ - (-x)_+$:

$$v_1(x + b_1)_+ + v_2(-x - b_1)_+ = (v_1 + v_2)(x + b_1)_+ - (v_1 + v_2)(x + b_1).$$

   The term $(v_1 + v_2)(x + b_1)$ can be absorbed into the residual connection, again reducing the representational cost of the function.

The residual connection allows us to conclude that for optimal networks solving (3), no two neurons activate at the same location. This provides a one-to-one correspondence between slope changes in the function and neurons in the network. The same conclusion also holds for any $T$ output ReLU network.

Therefore, any set of $T$ continuous piecewise linear (CPWL) functions with $K$ slope changes or knots, at locations $\tilde{x}_1, \ldots, \tilde{x}_K$ and slope changes $\mu_{kt}$ is represented, with minimal representational cost, by a network of width $K$, where parameters satisfy $-b_k/w_k = \tilde{x}_k$ and $\mu_{kt} = w_k v_{kt}$. This correspondence enables us to analyze networks which solve (3) entirely using CPWL functions, treating "knots" and "neurons" interchangeably and using $|v_{kt}|$ to denote both the magnitude of the slope change at a knot and the magnitude of the output weight.

**Connect-the-dots interpolation is always a solution to** (3)**.** Using Theorem 3.2, we proceed to prove Theorem 3.1. The objective function in (3) is coercive, moreover the interpolation constraint is a closed set due to the fact that neural networks are continuous with respect to their parameters. Therefore a solution to (3) exists (Beck, 2017, Lemma 2.14). Let $S_{\boldsymbol{\theta}}^*$ denote the set of parameters of optimal neural networks which solve (3) for the given data points, and let

$$S^* := \{f : \mathbb{R} \to \mathbb{R}^T \mid f(x) = f_{\boldsymbol{\theta}}(x) \ \forall x \in \mathbb{R}, \ \boldsymbol{\theta} \in S_{\boldsymbol{\theta}}^*\} \tag{26}$$

be the set of functions represented by neural network with optimal parameters in $S_{\boldsymbol{\theta}}^*$. First, note that the connect-the-dots interpolant $f_{\mathcal{D}}$ is in the solution set $S^*$. To see this, fix any $f \in S^*$, and apply Theorem 3.2 repeatedly to remove all "extraneous" knots (i.e., knots located away from the data points $x_1, \ldots, x_N$) from $f$. By Theorem 3.2, the resulting function—which is simply $f_{\mathcal{D}}$—has representational cost no greater than the original $f$, and since $f$ had optimal representational cost, so does $f_{\mathcal{D}}$.

**Conditions for non-unique solutions.** We first prove the conditions for which (3) admits an infinite number of solutions. For some $i = 2, \ldots, N - 2$, consider the two vectors

$$\boldsymbol{s}_i - \boldsymbol{s}_{i-1} = \frac{\boldsymbol{y}_{i+1} - \boldsymbol{y}_i}{x_{i+1} - x_i} - \frac{\boldsymbol{y}_i - \boldsymbol{y}_{i-1}}{x_i - x_{i-1}} \tag{27}$$

and

$$\boldsymbol{s}_{i+1} - \boldsymbol{s}_i = \frac{\boldsymbol{y}_{i+2} - \boldsymbol{y}_{i+1}}{x_{i+2} - x_{i+1}} - \frac{\boldsymbol{y}_{i+1} - \boldsymbol{y}_i}{x_{i+1} - x_i}. \tag{28}$$

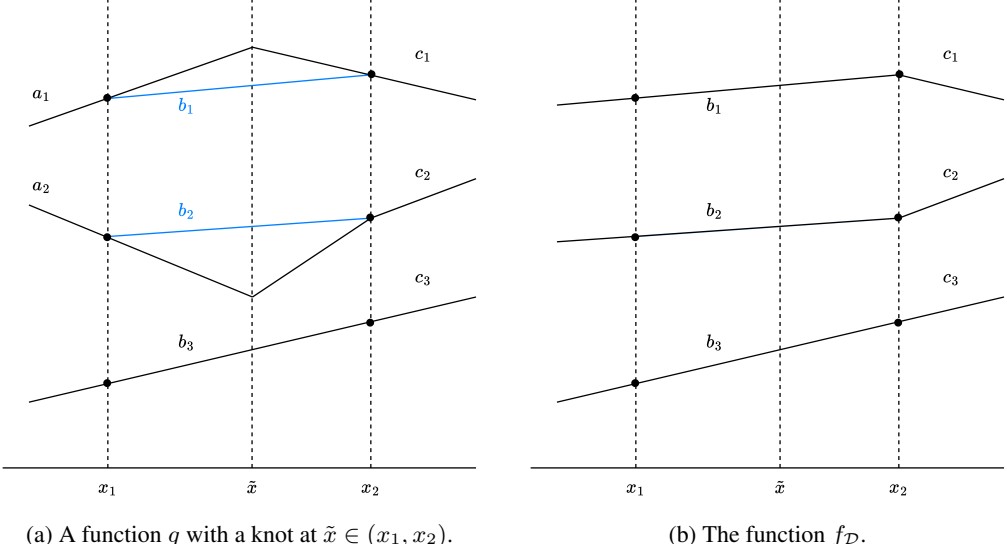

(a) A function $g$ with a knot at $\tilde{x} \in (x_1, x_2)$.  (b) The function $f_{\mathcal{D}}$.

Figure 5: Left: a function $g$ which has a knot in one or more of its outputs at a point $\tilde{x} \in (x_1, x_2)$. Right: the connect-the-dots interpolant $f_{\mathcal{D}}$. The representational cost of $g$ is strictly greater than that of $f_{\mathcal{D}}$.

and assume they are aligned. Then we may view the function $f_{\mathcal{D}}$ around the interval $[x_i, x_{i+1}]$ as an instance of Theorem 3.2 with $\boldsymbol{a} = \boldsymbol{s}_{i-1}$, $\boldsymbol{b} = \boldsymbol{s}_i$, and $\boldsymbol{c} = \boldsymbol{s}_{i+1}$. Fix some point $\tilde{x} \in (x_i, x_{i+1})$ and denote

$$\tau = \frac{\tilde{x} - x_i}{x_{i+1} - x_i}.$$

Let $\boldsymbol{\delta}$ be any vector which is aligned with $\boldsymbol{a} - \boldsymbol{b}$ and $\frac{1-\tau}{\tau}(\boldsymbol{b} - \boldsymbol{c})$ and has smaller norm than both, and let $f : \mathbb{R} \to \mathbb{R}^T$ be the function whose output slopes on $(x_i, \tilde{x})$ are given by $\boldsymbol{\delta}$ and whose slopes on $(\tilde{x}, x_{i+1})$ are given by $\boldsymbol{b} - \frac{\tau}{1-\tau}\boldsymbol{\delta}$. Then by Theorem 3.2, $R(f) = R(f_{\mathcal{D}})$ and thus $f \in S^*$. Since there are infinitely many $\boldsymbol{\delta}$'s which satisfy the condition above to choose the solution to (3) is non-unique in this case with infinitely many optimal solutions.

**Necessary and sufficient condition under which $f_{\mathcal{D}}$ is the unique solution.**  For the other direction of the proof, suppose that for any $i = 1, \ldots, N - 1$, the vectors $\boldsymbol{s}_i - \boldsymbol{s}_{i-1}$ and $\boldsymbol{s}_{i+1} - \boldsymbol{s}_i$ are *not* aligned, and assume by contradiction that there is some $f \in S^*$ which is *not* of the form $f_{\mathcal{D}}$. This $f$ must not have any knots on $(-\infty, x_1]$ or $[x_N, \infty)$, since removing such a knot would strictly decrease $R(f)$ without affecting the ability of $f$ to interpolate the data, contradicting optimality of $f$. So it must be the case that $f$ has an extraneous knot at some $\tilde{x}$ which lies between consecutive data points $x_i$ and $x_{i+1}$. Let $g$ denote the function obtained by removing all extraneous knots from $f$ *except* the one located at $\tilde{x}$. By Theorem 3.2, $R(g) \le R(f)$.

Now, suppose the extraneous knot is between $[x_i, x_{i+1}]$ for $i = 2, \ldots, N - 2$. Since $g$ has no extraneous knots away from $\tilde{x}$, it must be the case that $g$ agrees with $f_{\mathcal{D}}$ on $[x_{i-1}, x_i]$ and $[x_{i+1}, x_{i+2}]$. We may view the behavior of $g$ around the interval $[x_i, x_{i+1}]$ as an instance of Theorem 3.2 with $\boldsymbol{a} = \boldsymbol{s}_{i-1}, \boldsymbol{b} = \boldsymbol{s}_i$, and $\boldsymbol{c} = \boldsymbol{s}_{i+1}$. By assumption, $\boldsymbol{a} - \boldsymbol{b}$ and $\boldsymbol{b} - \boldsymbol{c}$ are *not* aligned, so by Theorem 3.2, removing the knot at $\tilde{x}$ would strictly reduce $R(g)$. This contradicts optimality of $g$, hence of $f$.

Finally, consider the case where the extraneous knot is on the interval $[x_i, x_{i+1}]$ where $i = 1$ (the case $i = N - 1$ follows by an analogous argument). Let $\boldsymbol{a}$ denote the vector of incoming slopes of $g$ at $x_1$. Define

$$\boldsymbol{b} = \frac{\boldsymbol{y}_2 - \boldsymbol{y}_1}{x_2 - x_1} \quad \boldsymbol{c} = \frac{\boldsymbol{y}_3 - \boldsymbol{y}_2}{x_3 - x_2}.$$

Since $g$ has no extraneous knots except for $\tilde{x}$, the slopes of $g$ coming out of $x_2$ are $\boldsymbol{c}$. By optimality of $g$, it must be the case that $\boldsymbol{a} - \boldsymbol{b}$ and $\boldsymbol{b} - \boldsymbol{c}$ are aligned (otherwise we could invoke Theorem 3.2 and strictly reduce the representational cost of $f$ by removing the knot at $\tilde{x}$, a contradiction), which implies that $\mathrm{sgn}(a_t - b_t) = \mathrm{sgn}(b_t - c_t)$ for each $t = 1, \ldots, T$. For any outputs $t$ which have a knot

at $\tilde{x}$, this quantity is nonzero, in which case $|c_t - b_t| < |c_t - a_t|$ (see Fig. 5a). Let $1, \ldots, t_0$ be the indices of the outputs which have a knot at $\tilde{x}$, and let $t_0 + 1, \ldots, T$ be the indices of the outputs which do *not* have a knot at $\tilde{x}$. We may again invoke Theorem 3.2 to remove the knots from $g$, resulting in a new function $\tilde{g}$ (satisfying $R(g) \geq R(\tilde{g})$) which has slopes $\boldsymbol{a}$ coming into $x_1$, $\boldsymbol{b}$ between $x_1$ and $x_2$, and $\boldsymbol{c}$ coming out of $x_2$. The contribution of these knots to $R(\tilde{g})$ is then given by:

$$
\left\| \begin{bmatrix} b_1 - a_1 \\ \vdots \\ b_{t_0} - a_{t_0} \end{bmatrix} \right\|_2 + \left\| \begin{bmatrix} c_1 - b_1 \\ \vdots \\ c_{t_0} - b_{t_0} \end{bmatrix} \right\|_2 \geq \left\| \begin{bmatrix} b_1 - a_1 + c_1 - b_1 \\ \vdots \\ b_{t_0} - a_{t_0} + c_{t_0} - b_{t_0} \end{bmatrix} \right\|_2
$$

$$
= \left\| \begin{bmatrix} c_1 - a_1 \\ \vdots \\ c_{t_0} - a_{t_0} \end{bmatrix} \right\|_2
$$

$$
> \left\| \begin{bmatrix} c_1 - b_1 \\ \vdots \\ c_{t_0} - b_{t_0} \end{bmatrix} \right\|_2
$$

but the last quantity is exactly the contribution of these knots to $R(f_{\mathcal{D}})$ (see Fig. 5b). This contradicts optimality of $\tilde{g}$, hence of $g$ and of $f$. The remainder of the proof is dedicating to showing that such datasets which admit non-unique solutions are rare when the data is randomly sampled from a continuous distribution.

**Datasets which admit non-unique solutions have Lebesgue measure zero.**  If $N = 2$ or $N = 3$, then $f_{\mathcal{D}}$ is the only solution to (3), so we focus on the case where $N \geq 4$. Suppose that for some $i \in \{2, \ldots, N - 2\}$, the data points $x_{i-1}, x_i, x_{i+1}, x_{i+2} \in \mathbb{R}$ and labels $\boldsymbol{y}_{i-1}, \boldsymbol{y}_i, \boldsymbol{y}_{i+1}, \boldsymbol{y}_{i+2} \in \mathbb{R}^T$ satisfy the requirement that

$$
\frac{\boldsymbol{y}_{i+1} - \boldsymbol{y}_i}{x_{i+1} - x_i} - \frac{\boldsymbol{y}_i - \boldsymbol{y}_{i-1}}{x_i - x_{i-1}} = w \left( \frac{\boldsymbol{y}_{i+2} - \boldsymbol{y}_{i+1}}{x_{i+2} - x_{i+1}} - \frac{\boldsymbol{y}_{i+1} - \boldsymbol{y}_i}{x_{i+1} - x_i} \right) \tag{29}
$$

for some $w > 0$, where both vectors are nonzero. After some computation, this is equivalent to the requirement that

$$
(x_{i+1} - x_i)\boldsymbol{y}_{i-1} - w(x_{i+2} - x_{i+1})\boldsymbol{y}_i + \big((1 - w)x_i - x_{i+1} + wx_{i+2}\big)\boldsymbol{y}_{i+1} - w(x_{i+1} - x_i)\boldsymbol{y}_{i+2} = \boldsymbol{0} \tag{30}
$$

or equivalently

$$
\underbrace{[\boldsymbol{y}_{i-1}, \boldsymbol{y}_i, \boldsymbol{y}_{i+1}, \boldsymbol{y}_{i+2}]}_{\boldsymbol{Y}_i \in \mathbb{R}^{T \times 4}} \left( \underbrace{\begin{bmatrix} x_{i+1} - x_i \\ 0 \\ x_i - x_{i+1} \\ 0 \end{bmatrix}}_{\boldsymbol{a}_1 \in \mathbb{R}^4} - w \underbrace{\begin{bmatrix} 0 \\ x_{i+2} - x_{i+1} \\ -x_{i+2} \\ x_{i+1} - x_i \end{bmatrix}}_{\boldsymbol{a}_2 \in \mathbb{R}^4} \right) = \boldsymbol{0} \tag{31}
$$

for some $w > 0$. In order for this requirement to be satisfied, it must be the case that $\boldsymbol{Y}\boldsymbol{a}_1 = w\boldsymbol{Y}\boldsymbol{a}_2$ for some $w > 0$, or equivalently, that the matrix $\boldsymbol{U} = \boldsymbol{Y}[\boldsymbol{a}_1, \boldsymbol{a}_2] \in \mathbb{R}^{T \times 2}$ has rank one. Since the rank of any matrix and its Gram matrix are equivalent, this is equivalent to requiring that $\boldsymbol{U}\boldsymbol{U}^\top \in \mathbb{R}^{T \times T}$ has rank one, or equivalently (because $T > 1$), that $\det(\boldsymbol{U}\boldsymbol{U}^\top) = 0$. Now observe that, based on the definition of the determinant and of the matrix $\boldsymbol{U}$, the function $\det(\boldsymbol{U}\boldsymbol{U}^\top)$ is a real-valued polynomial function of the variables $\boldsymbol{Y}_i = [\boldsymbol{y}_{i-1}, \boldsymbol{y}_i, \boldsymbol{y}_{i+1}, \boldsymbol{y}_{i+2}] \in \mathbb{R}^{T \times 4}$ and $x_{i-1}, x_i, x_{i+1}, x_{i+2} \in \mathbb{R}^4$. Therefore, as the zero set of a polynomial function (i.e. an algebraic variety), the set of all $\boldsymbol{Y}_i = [\boldsymbol{y}_{i-1}, \boldsymbol{y}_i, \boldsymbol{y}_{i+1}, \boldsymbol{y}_{i+2}] \in \mathbb{R}^{T \times 4}$ and $x_{i-1}, x_i, x_{i+1}, x_{i+2} \in \mathbb{R}^4$ for which $\det(\boldsymbol{U}\boldsymbol{U}^\top) = 0$ has Lebesgue measure zero. This is true for any $i = 1, \ldots, N$, so taking the union over $i = 1, \ldots, N$, we have that the set of all possible data points $x_1, \ldots, x_N \in \mathbb{R}$ and label vectors $\boldsymbol{y}_1, \ldots, \boldsymbol{y}_N \in \mathbb{R}^T$ which admit non-unique solutions has Lebesgue measure zero (as a subset of $\mathbb{R}^N \times \mathbb{R}^{T \times N}$). $\qquad \square$

# B  Additional Experimental Results and Details

## B.1  Numerical Illustration of Theorem 3.1

We provide numerical examples to illustrate the difference in solutions obtained from single task versus multi-task training and validate our theorem. The first row in Fig. 6 shows three randomly initialized univariate neural networks trained to interpolate the five red points with minimum sum of squared weights. While all three of the learned functions have the same representational cost (i.e., all minimize the second-order total variation subject to the interpolation constraint), they each learn different interpolants. This demonstrates that gradient descent does not induce a bias towards a particular solution. The second row shows the function learned for the first output of a multi-task neural network. This network was trained on two tasks. The first task consists of interpolating the five red points while the second consists of interpolating five randomly generated labels sampled from a standard Gaussian distribution. When trained to interpolate with minimum sum of squared weights we see that the connect-the-dots solution is the only solution learned regardless of initialization, verifying Theorem 3.1. This solution simultaneously minimizes the second-order total variation and (on the interval $[x_1, x_N]$ the norm in the first-order Sobolev RKHS $H^1([x_1, x_N])$ associated with the kernel $k(x, x') = 1 - (x - x')_+ + (x - x_1)_+ + (x_1 - x')_+$ De Boor and Lynch (1966), subject to the interpolation constraints. [5]

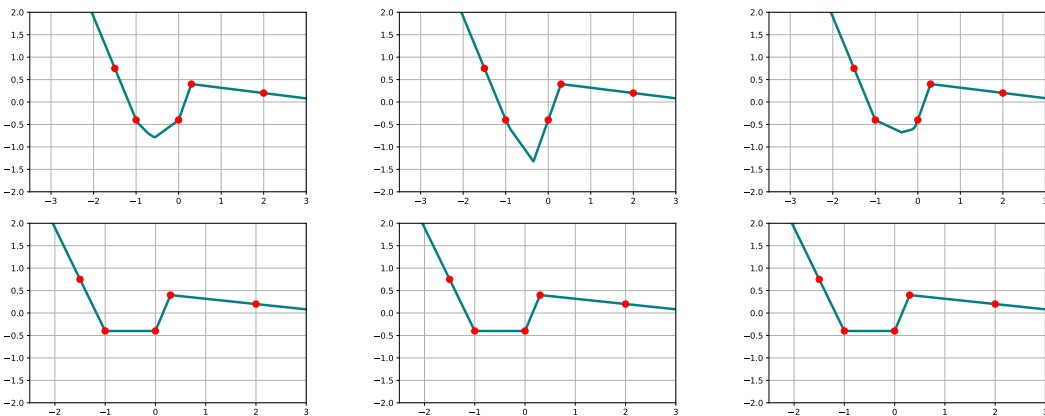

Figure 6: *Top Row*: Three randomly initialized neural networks trained to interpolate the five red points with minimum sum of squared weights. *Bottom Row*: Three randomly initialized two-output neural networks trained to interpolate a multi-task dataset with minimum sum of squared weights. The labels for the first task are the five red points shown while the labels for the second were randomly randomly sampled from a standard Gaussian distribution.

## B.2  Additional Experiments from Section 4

All of our experiments were carried out in PyTorch and used the Adam optimizer. We trained the models with mean squared error loss and included the representational cost $\sum_{k=1}^{K} \|\boldsymbol{v}_k\|_2$ as a regularizer with $\lambda = 1e - 5$ for the univariate experiments and $\lambda = 1e - 3$ for the multi-variate experiments. All models were trained to convergence. The networks were initialized with 20 neurons for the univariate experiments and 800 neurons for the multi-variate experiments. For solving (25) we utilized CVXPy Diamond and Boyd (2016).

Fig. 7 below provides additional experimental results to accompany the discussion in Section 4. The results demonstrate that our observations from Section 4 are true setting across multiple random initializations of the network.

---

[5]The code for reproducing all numerical experiments can be found at `https://github.com/joeshenouda/effects-mtl-nns`.

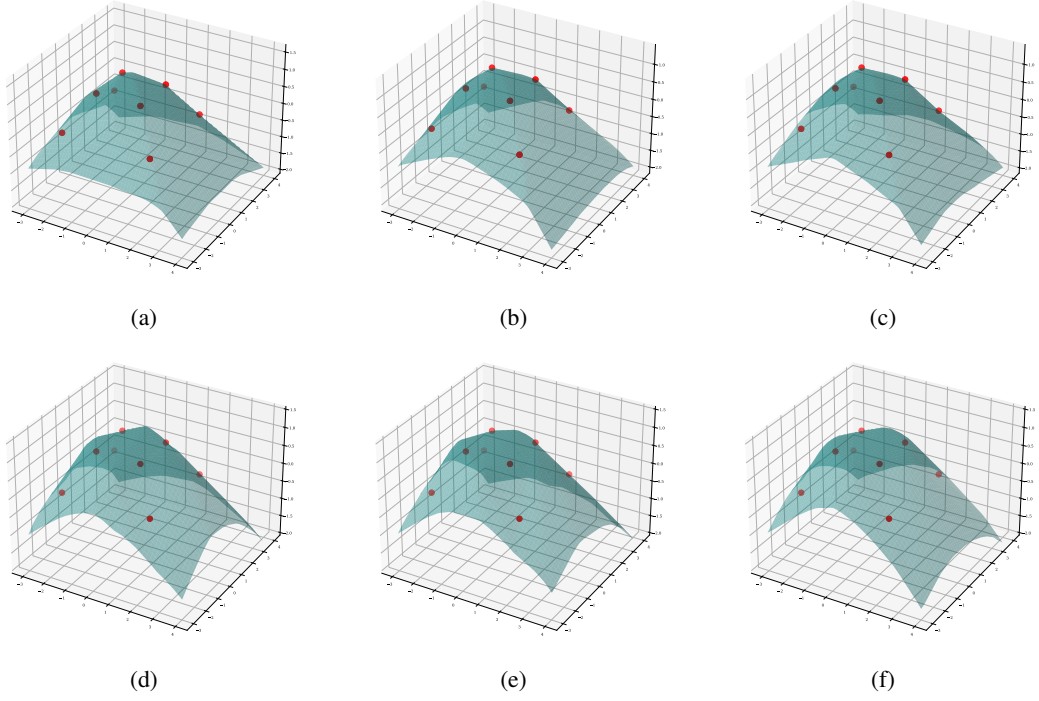

| (a) | (b) | (c) |
| (d) | (e) | (f) |

Figure 7: We present three more trials of the same experiment from Section 4. The top row corresponds to the solution of the fist output of a multi-task neural network with $T = 101$ tasks. The first task is the original (i.e. interpolating the red points), the other $100$ are randomly sampled i.i.d from a Bernoulli distribution with equal probability for one and zero. The second row corresponds to the solutions obtained by solving (25). We see again that for the $T = 101$ multi-task neural network the learned function is consistent across multiple random initializations. Moreover, those solutions are also similar to the ones obtained by solving (25). These results suggest that with many diverse tasks the contributions of any one task on the optimal neurons are not significant.

### B.3 High Dimensional Multivariate Experiments

In this section we provide additional experiments in a higher dimensional setting to demonstrate how multi-task solutions can differ drastically from single-task. For these experiments we consider a student-teacher model. In particular, we generated $25$ random ReLU neurons with unit norm input weights $\boldsymbol{w}_t \in \mathbb{R}^5$ for $t = 1, \ldots, 25$. These served as "teacher" neurons. We then generated a multi-task dataset $\{\boldsymbol{x}_i, \boldsymbol{y}_i\}_{i=1}^{20}$ where each $\boldsymbol{x}_i \in \mathbb{R}^5$ and sampled i.i.d from a standard multi-variate Gaussian distribution. The labels $\boldsymbol{y}_i \in \mathbb{R}^{25}$ were then generated according to the teacher ReLU neurons, that is,

$$y_{it} = (\boldsymbol{w}_t^T \boldsymbol{x}_i)_+.$$

Therefore, each task is generated by a single-index model. We then trained $25$ student single-output ReLU neural networks on each tasks as well as a 25-output multi-task ReLU neural network on all the tasks. Both were trained to minimize MSE loss and were regularized a weight decay parameter of $\lambda = 1e - 4$. All networks nearly interpolated the data with MSE value less than $1e - 4$. Figure 10 plots the learned single task networks evaluated along a a unit norm vector $\boldsymbol{w} \in \mathbb{R}^5$. From the plots it is clear that the single task networks recover the ground truth function (i.e. a single ReLU neuron) as it looks like a ReLU ridge function in every direction. Moreover, we observed a an average sparsity of five active neurons across all the trained single-output networks.

In the Figure 9 we also plot the output of the $t^{\text{th}}$ function from the leanred multi-task network evaluated at the same $\boldsymbol{w}$. In this case, the functions look very different from a single-index model and do not recover the ground truth data-generating function for the respective task. Figure 8 shows

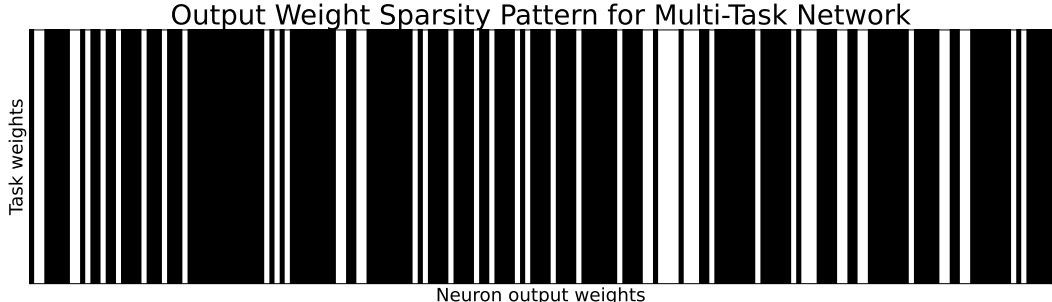

Figure 8: Sparsity pattern for output weight matrix of the multi-task student network. The $k^{\text{th}}$ column in the matrix corresponds to the output weight of the $k^{\text{th}}$ neuron. We observe that each neuron either contributes to all the tasks or none.

the sparsity pattern of the weights for each neuron with roughly $150$ neurons contributing to all the outputs.

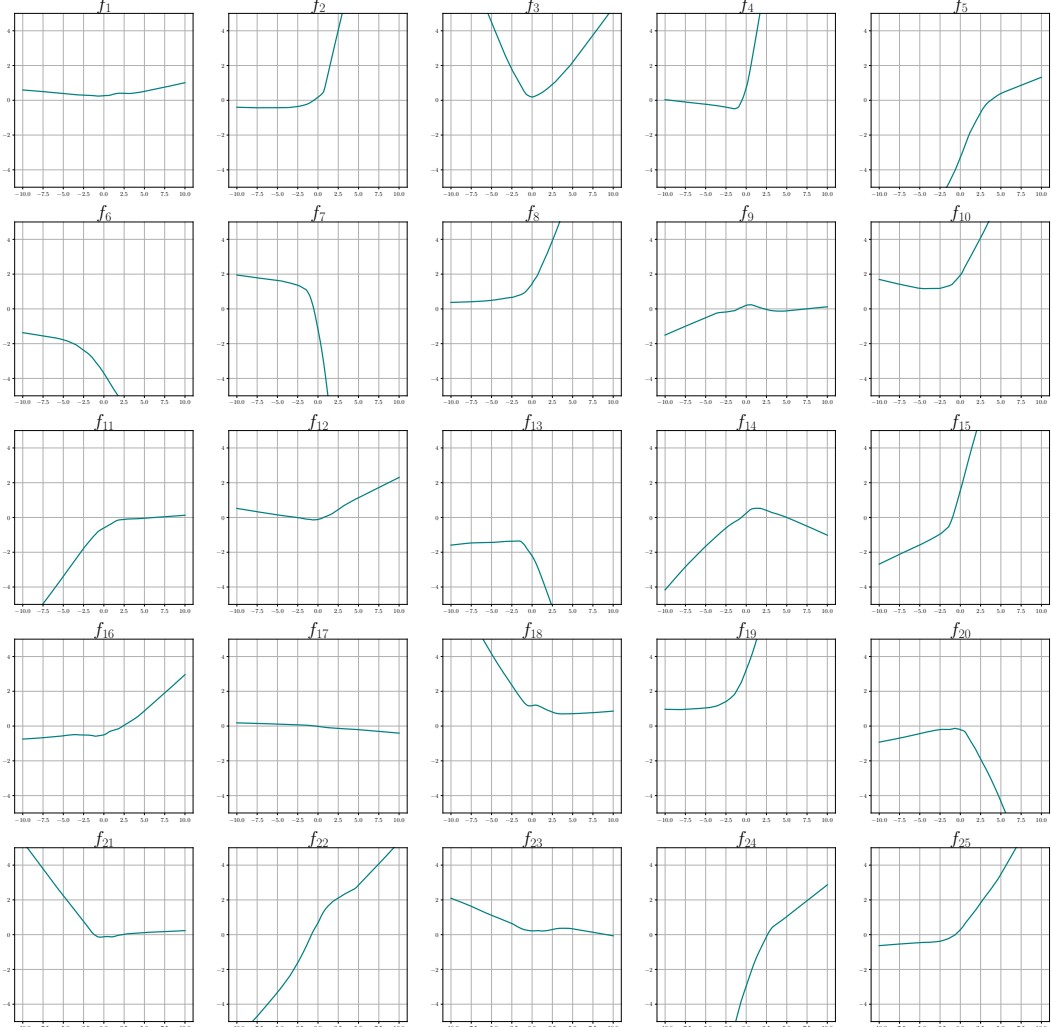

Figure 9: Multi-task solutions along the same direction $w$. Here $f_t$ denotes the $t^{\text{th}}$ output of the multi-task network. We observe that unlike Figure 10 the functions do not look like ReLU ridge functions and have variation in multiple directions.

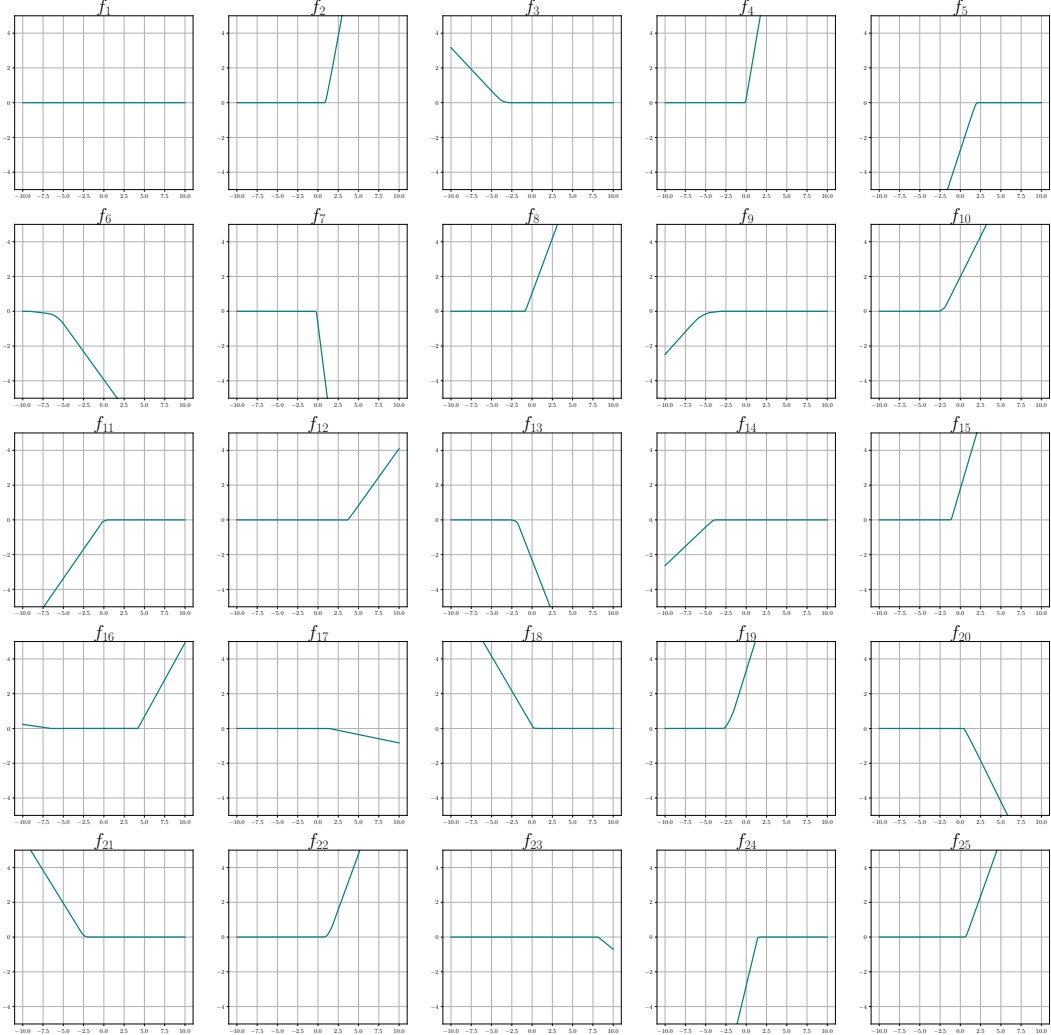

Figure 10: The 25 single-task networks evaluated along the same direction $w$ as in Figure 9. Here $f_t$ denotes the $t^{\text{th}}$ single-task network trained on task $t$ according to the data generating function above. Here as we expect the single-task nets are ReLU ridge functions. We note that these observations hold across different choices of the one-dimensional subspace $w$.

