# OpenReview forum: "A New Neural Kernel Regime: The Inductive Bias of Multi-Task Learning"
_NeurIPS.cc/2024/Conference — NeurIPS 2024 poster_

### Official Review · Reviewer_aewJ · 2024-06-14

**Soundness:** 3
**Presentation:** 3
**Contribution:** 3
**Rating:** 6
**Confidence:** 4

**Summary:**

This paper analyzes the solutions to a multi-task training objective that entails finding a two-layer ReLU network that interpolates all training data and has minimum l2 norm of the first- and second-layer weights, in the overparameterized setting in which the number of neurons is larger than the total number of training examples across tasks. In the case wherein the the data dimension is one (univariate case) and under a very weak task diversity condition, the paper proves that the solution to the multi-task training objective is unique and equal to the ``connect-the-dots’’ interpolator, which aligns with the solution to a particular kernel optimization problem with a certain kernel. On the other hand, prior results, as well as empirical results presented in this paper, show that the solutions to the analogous single-task training objective are not unique and may be overly-complicated, non-robust interpolators. The paper takes an initial step towards extending the univariate multi-task learning results to the multivariate setting with preliminary analysis and experiments suggesting that the multi-task solution is unique, unlike the single-task solutions.

**Strengths:**

- The writing is mostly clear and easy to follow.
- The univariate result is interesting. Such a clear separation between the behavior of single-task and multi-task learning, with only a very weak assumption on the tasks, is arguably unprecedented in the literature to my knowledge, and is an important step towards explaining why multi-tasking can lead to more robust models. I checked the proof and did not notice any mistakes.
- The paper studies a nonlinear model, whereas most existing characterizations of solutions to multi-task objectives are limited to linear models.
- In the multivariate case, the provided intuitions are sound, and the experiments evince the validity of the approximations in the discussion and the uniqueness of solution.

**Weaknesses:**

- The most interesting results are limited to the univariate case. In the multivariate case, the analysis is not rigorous. Also, the conclusion that multi-tasking behaves like a kernel method is underwhelming because this kernel is unknown and depends on $\mathbf{v}^*$ and $\mathbf{w}_k^*$ and $b_k^*$ for $k=2,…,K$, and characterizing these reduces to the original problem of characterizing the solution(s) of the multivariate multi-task learning problem. Along this line, while the multivariate experiments are helpful for the reasons discussed above, they don’t help to characterize the kernel, and in particular they don’t show an analogous solution to the connect-the-dots solution for the univariate case.
- In addition to being primarily focused on the univariate case, the results are limited to the setting in which the overparameterized networks are trained to exactly fit the training data, and all tasks have exactly the same input data, neither of which may be the case in practice.
- More discussion could be given regarding why the fact that multi-task learning behaves like a kernel method is important/interesting, and why the kernel in the univariate case is conducive to generalization.
- The optimization algorithm used in the experiments is not discussed. If the algorithm is gradient descent on the Lagrangian, then this suggests that the multi-task augmented loss is convex, and the single-task augmented loss is non-convex, which is possibly an important point that the paper is missing. Related, for the experiments showing that single-tasking with different initializations can lead to different solutions, it would be helpful to share the final values of the training objective, to confirm that all functions are global optima.

Minor
- Nit-picking, but the use of “unlikely” to describe the alignment of $\mathbf{s}_i - \mathbf{s}_{i-1}$ and $\mathbf{s}_{I+1} - \mathbf{s}_{I}$ is not quite correct because there is no assumed generative model for the data. Under many distributions, such alignment is very likely.
- Related: I suggest that the authors refer to the condition that $\mathbf{s}_i - \mathbf{s}_{i-1}$ and $\mathbf{s}_{I+1} - \mathbf{s}_{I}$ are not aligned as a task diversity condition. If all the tasks are the same, it fails (which is an important sanity check that should be pointed out).  However, if the tasks have only a small amount of diversity in some sense, it holds. Previous studies of multi-task learning require some level of task diversity to learn generalizable models, e.g. [1,2,3,4].
- Again, related:  there is a large body of work studying solutions found by multi-task learning and their generalization capabilities [1,2,3,4, etc.] that the paper does not address. Like the current paper, these works also require task diversity, so they are more related than the paper’s brief discussion suggests.
- Citations should be changed to parenthetical.

**Questions:**

N/A

**Limitations:**

The authors have adequately addressed limitations.

---

> ### Author Rebuttal · Authors · 2024-08-07
>
> We thank the reviewer for their careful review of our work and the concerns brought up. Below, we address each of the weaknesses brought up in a point-by-point manner.
>
> 1. **[Multi-variate case]** The results for the multivariate case (which also apply to the univariate setting) are more of an approximate argument.  This is partially due to the fact that far less is known about the solution sets in the multivariate case (see e.g. [3],[4], [8]) compared the the complete characterization which is known for univariate solutions ([2]). The fact that multivariate solutions can also display variety in orientation, as evidenced in the example of Figure 5, hint at the potential richness of the solution sets. Under the iid task model used in our analysis, we believe that the approximations assumed in the analysis can be made more rigorous, which is of interest for future work. We disagree that the results are underwhelming. The fact that multi-task neural network training behaves like an $\ell^2$ minimization over linear combinations of the learned neurons, whereas single-task training is equivalent to $\ell^1$ minimization over the neurons, implies that the nature and properties of functions the functions learned by multi-task vs. single-task training can be profoundly different. This is, in our opinion, novel and interesting.
>
> 2. **[Interpolation]** We note that the more approximate argument presented in the multivariate case (which also applies to the univariate case) characterizes non-interpolation training problems which minimize a data-fitting term plus a penalized loss (for any penalty strength). Neural network interpolation problems are also a setting of practical interest, since many overparameterized networks trained in practice are able to achieve zero or near-zero training error. Moreover, standard multi-task learning scenarios generally assume that all tasks share the same set of input data points, and these tasks are learned jointly, which is exactly the setup we consider.``
>
>
> 3. **[Relevance of kernel regime]** The fact that multi-task learning behaves like a kernel method is important/interesting because it shows that the solutions to each task can be profoundly different compared to those obtained by training a separate network for each, even if the tasks are statistically independent as assumed in our analysis. In other words, multi-task training can have a major effect on the learned functions for each task even in situations where one might not anticipate such an effect. This insight may be valuable in practice, since this effect of multi-task training may be desirable or undesirable depending on the application. Regarding generalization properties, see the general comment made to all reviewers above.
>
>
> 4. **[Non-convex objective]** As discussed in Appendix 8, all experiments except that in the bottom right of Fig. 5 are done by training multi-output neural networks using the Adam optimizer with MSE loss and weight decay parameter 1e-5 (univariate experiments) or 1e-3 (multivariate experiments). Because the data-fitting term is non-convex and lambda is very small, the resulting problem is always non-convex. For the experiments in Fig. 4 showing that multi-task training with different random initializations leads to the same solution, it is not necessary to check the objective values since we know a priori (by our result in Theorem 3.1) that the learned functions depicted in the figures are global minimizers. However, if it is helpful to the presentation, we can include these numbers. For the multi-variate experiments in Fig. 5 the global optimality of these solutions were verified using the convex neural networks approach in Tolga & Pilanci [8].
>
> Regarding the minor weaknesses:
>
> 1. We agree that the use of the word "unlikely" here is imprecise since no distributional assumptions are given on the data/labels. We note that Corollary 1 (Appendix 7.2) of Theorem 3.1 implies that, if the task labels are sampled i.i.d. from an absolutely continuous distribution (e.g. Gaussian) and the number of tasks T is greater than 1, then those labels admit only the connect-the-dots solution with probability one. (This follows from Corollary 1 in Appendix 7.2.)  So for example, if the labels are given by ground truth values plus some additive Gaussian noise—even a small amount of noise—the unique multi-task solution is connect-the-dots. We will mention this explicitly and update the phrasing to be more precise.
> 2. Agreed.
> 3. We are unsure of what the reviewer is referring to for citations [1,2,3,4] but we can include some more discussion/mention of existing multi-task learning research which focuses on the benefits of unrelated tasks, and how our work relates to these upon the reviewers clarification.
> 4. Agreed.
>
> [2] Hanin, Boris. "On The Implicit Bias of Weight Decay in Shallow Univariate ReLU Networks."
>
> [3] Ongie, Greg, et al. "A Function Space View of Bounded Norm Infinite Width ReLU Nets: The Multivariate Case" ICLR 2020
>
> [4] Zeno, Chen, et al. "How do minimum-norm shallow denoisers look in function space?." NeurIPS 2024.
>
> [8] Pilanci, Mert, and Tolga Ergen. "Neural networks are convex regularizers: Exact polynomial-time convex optimization formulations for two-layer networks." ICML 2020.

---

> > ### Comment · Reviewer_aewJ · 2024-08-11
> >
> > I thank the authors for their thorough response, which has helped to alleviate my concerns, and I have increased my score as a result. I apologize for forgetting to include the citations, which are as follows:
> >
> > [1] Tripuraneni et al., On the theory of transfer learning: The importance of task diversity, NeurIPS 2020.
> >
> > [2] Du et al., Few-shot learning via learning the representation, provably, ICLR 2020.
> >
> > [3] Sun et al., Towards sample-efficient overparameterized meta-learning, NeurIPS 2021.
> >
> > [4] Collins et al., Provable Multi-Task Representation Learning by Two-Layer ReLU Neural Networks, ICML 2024

---

> > > ### Author Response · Authors · 2024-08-11
> > >
> > > We appreciate your response and thank you for increasing the score!
> > > Indeed, the works mentioned are very relevant and we will be sure to reference them in the camera-ready version of the paper.

---

### Official Review · Reviewer_XF9Q · 2024-06-21

**Soundness:** 2
**Presentation:** 3
**Contribution:** 2
**Rating:** 5
**Confidence:** 3

**Summary:**

The paper shows that multi-task training can benefit the single tasks, even if the tasks are unrelated. Assuming a particular setting of training a 2-layer neural network that finds a path norm interpolation solution on univariate data, the multi-task optimizer is unique and is given by the piecewise linear function that connects the training data points which coincides the minimum norm interpolant in a first-order Sobolev RKHS, whereas the corresponding single-task optimizer is non-unique and hence not the minimum norm interpolant with respect to any RKHS.

**Strengths:**

The fact that a multi-task learning objective can induce a kernel regime, in cases where the single-task setting is non-unique, is to the best of my knowledge novel and interesting. The solution and presentation are clear and simple and correspond to the piecewise linear function that interpolates the training data. The theoretical argument that motivates the relevance for multi-dimensional covariates is solid and empirically verified by a minimal experiment. The related work is adequately cited.

**Weaknesses:**

While investigating the benefits of multi-task settings has great potential, I cannot recommend the paper for acceptance. The points and questions below specify important limitations that remain to be addressed and essential aspects of multi-task training that were not explored:

- Question 1 below raises a potential gap in the proof.
- A general claim that the multi-task solution improves upon the single-task solution - an essential narrative in the paper - can clearly not be made, and depends on the exact functional dependence between covariates x and labels y. Canatar et al. (2021) is one example of how the generalization error of (other) neural kernel regimes depends on the task alignment. In the provided example in Figure 5, the ground truth function is arguably ambiguous and it might also be desirable to reflect this uncertainty by being able to learn several reasonable solutions as the single-task solution does. At least a more detailed empirical evaluation in varying input dimension, varying task signal strength and alignment could have yielded valuable insights in the regard of uniqueness and generalization in various settings. Even under negative outcome, the results would have yielded important information about the practical implications and limitations of the univariate theory.
- The implications and limitations for practical settings are unclear due to a lack of empirical evaluations. The experiments only include one task with signal and otherwise pure noise tasks (see questions 2 and 3 below), and at most cover 2 input dimensions with constructed symmetric data points.

**Typos:** in lines 26, 146, 222, before 235: be be, eq (3): f_theta(x_i)=y_i

**References:**
Canatar, A., Bordelon, B. & Pehlevan, C. Spectral bias and task-model alignment explain generalization in kernel regression and infinitely wide neural networks. *Nat Commun* **12**, 2914 (2021).

**Questions:**

- Assume an optimal solution does not contain a training point x_i as a knot. How can you transform this solution to the linear spline containing only the x_i as knots. Do all optimizers already contain all x_i as knots? Can x_i be introduced without increasing the representational cost? I do not see how this case is covered in the proof and would appreciate a clarification.
- Is there an intuitive explanation when and why the multi-task solution improves the individual tasks too? In particular, does your example with all tasks except one being pure noise tasks induce some regularizing implicit bias or does this conclusion also hold for more typical multi-task settings where there is a significant amount of signal in each task, and differing amounts of alignment between them?
- Practical data sets are not expected to be as symmetrical as the x_i in the 2-dimensional example of Figure 5. Is the non-uniqueness of single-task solutions less severe under continuous iid sampling of the x_i?
- The results of Boursier and Flammarion (2023) indicate that regularizing the biases can make an important difference in the optimal solutions. Could your multi-task theory be extended to this case and would the solution also be sparsified compared to the current objective without bias regularization?

**Limitations:**

The implications and limitations to practically optimized neural networks, multi-dimensional data and tasks with varying degrees of alignment and anti-alignment should have been assessed, at least empirically.

---

> ### Author Rebuttal · Authors · 2024-08-06
>
> We thank the reviewer for their careful review of our work and for their feedback. Below, we address each of the comments individually.
>
> 1. **[Clarification on Theorem 3.1]** We thank the review for this clarifying question about Theorem 3.1. Lemma 3.2 says that, given any solution which interpolates the data, we can remove all knots located away from the data points ("extraneous knots") without increasing the representational cost of the network. Removing these "extraneous knots" may introduce new knots at the data points which were not originally present, but Lemma 3.2 says that the total cost $R(0) = \|b-a\|_2 + \|c-b\|_2$ of the new solution  with knots at the data points (see Fig. 3) is no greater than the cost $R(\delta) = \| \delta + b-a \|_2 + \frac{1}{1-\tau} \| \delta \|_2 + \| c-b + \tau \delta/(1-\tau) \|_2$ of the original solution, which may or may not have knots at the data points. Therefore, the question does not present a gap in the proof. Please let us know if this clarifies the concern.
> 2.  **[Generalization]** See the general response to all reviewers above. When we referenced generalization, we specifically meant to reference the "tempered overfitting" result in [3], which is specific to the univariate case.
> 3. **[Experiments]** See the general response for details on new experimental results.
>
> Regarding questions:
>
> 1. We address this question in the rebuttal to weaknesses above.
> 2.  In the univariate case, the intuition is that multi-output weight decay regularization encourages neuron sharing across outputs ([6]) while also encouraging the weights of each output individually to be "small," i.e., encouraging each output individually to be a min-norm univariate interpolant (which are characterized in [2]). In general, the only way to accomplish this is for each output to perform connect-the-dots interpolation, with the shared knots located at the shared data points. The multivariate case is more subtle. The example in Figure 5 was constructed specifically to illustrate the existence of multiple solutions with signficantly different "directions" of variation. We agree that it is difficult to argue that any particular one of these solutions is better, but the more symmetric solution obtained by multi-task training does stay closer on average to the data points (we verified this computationally). We used pure noise tasks here for illustration and convenience, but similar results are obtained if the multiple tasks are randomly perturbed versions of the two-squares task. Intuitively and more generally, since the multi-task solutions are approximately equal to the kernel solution, these functions will be weighted combinations of all neurons in the model and rarely strongly aligned with a subset of the neurons.  Thus, the multi-task solutions tend to be smoother and have more symmetry.
> 5. In the new experiments conducted we do not rely on any special structure for the data samples. In this case we still observe a striking difference between solutions learned by multi-task task networks vs. single task networks trained individually.
> 6. This is an interesting theoretical question for future work. Our initial experiments (not shown) indicate a similar kernel-like effect when regularizing biases in multi-task settings.
>
> [2] Hanin, Boris. "On The Implicit Bias of Weight Decay in Shallow Univariate ReLU Networks."
>
> [6] Shenouda, Joseph, et al. "Variation Spaces for Multi-Output Neural Networks: Insights on Multi-Task Learning and Network Compression." JMLR 2024.

---

> > ### Comment · Reviewer_XF9Q · 2024-08-12
> >
> > Thank you for clarifying my confusion with the proof. I am still unable to find a statement anywhere in the proof that discusses introducing knots and why this does not increase the representational cost. I encourage the authors to clarify that in the proof.
> >
> > In the added experiments, it is not surprising that the single-task and multi-task solutions differ; the question is how. The fact that more neurons are active and they are shared across tasks is an interesting mechanistic understanding in this regard, but I would like to see the correspondence to Figure 1 in the attached pdf of single task NNs to have a comparison. Also, in Figure 2, it now seems that the single-task solutions are the piecewise linear interpolants and without variation across random seeds, which questions the generalization of the original interpretation in the paper. In this example, the multi-task solution does not seem to be desirable but instead further away from the piecewise linear interpolant of the data than the single task solution. It also remains unclear how task alignment or high dimensionality impacts the conclusions drawn in the paper.
> >
> > Overall, I will keep my score.

---

> > > ### Author Response · Authors · 2024-08-12
> > > **Follow-up to reviewer XF9Q**
> > >
> > > We thank the reviewer for engaging in the rebuttal. We want to further clarify that we are only considering interpolating solutions for the univariate setting. Therefore, we only consider solutions which pass through the data points exactly: it is never necessary to introduce additional knots in order to have the function interpolate the data points. However, if we consider an interpolating solution which has some "extraneous knots" (knots which are not located at the data points) and we remove these, additional knots may appear at the data points. Lemma 3.2 says that doing this will never increase the representational cost, and almost always decrease it, even if doing so results in new knots appearing at the data points. While we believe that this is clear in the current statement of the lemma, we are happy to update our language to make this more explicit.
> > >
> > > Regarding the new experiments, we would like to clear up some possible misunderstandings about our setup and claims.
> > >
> > > > In the added experiments, it is not surprising that the single-task and multi-task solutions differ;
> > >
> > > We disagree with this claim. In our experiment each task is entirely independent of the rest so there is no reason to assume that the network would exploit some shared representation. The network could have learned the ground truth input weight for each task (a single neuron) and then learned a diagonal output weight matrix for the output weight of each of those teacher tasks. Due to the unrelatedness of the different tasks this would seem like a reasonable solution however it is not the solution ultimately learned.
> > >
> > > > the question is how.
> > >
> > > For the univariate case **we precisely describe how the solutions are different**. Training a single task network can lead to many solutions that interpolate with minimum representation cost. However, doing multitask training almost always leads to the unique “connect-the-dots” interpolant for each task. Moreover, this interpolant is also the solution to minimum norm data interpolation in a particular RKHS for each task individually and we explicitly provide the kernel associated with this RKHS.
> > >
> > > The results for the multivariate case (which also apply to the univariate setting) are more of an approximate argument. This is partially due to the fact that far less is known about the solution sets in the multivariate case (see e.g. [1],[2], [3]) compared to the complete characterization which is known for univariate solutions ([4]). However, **our analysis and experiments do provide insights into the difference between single-task and multi-task solutions**. Multi-task neural networks learn solutions that behave like an $\ell_2$
> > >  minimization over linear combinations of the learned neurons, whereas single-task neural networks learn solutions that  behave as an $\ell_1$
> > > minimization over linear combinations the learned neurons. This is, in our opinion, novel and interesting and as pointed out by reiviewer `aewJ` is **arguably unprecedented in the literature.**
> > >
> > > >  but I would like to see the correspondence to Figure 1 in the attached pdf of single task NNs to have a comparison
> > >
> > > As mentioned in the experimental details in the general response, for the single task networks there are only 5-10 active neurons at the end of training.
> > >
> > > > Also, in Figure 2, it now seems that the single-task solutions are the piecewise linear interpolants and without variation across random seeds, which questions the generalization of the original interpretation in the paper.
> > >
> > > In this case the ground truth tasks are single ReLU neurons and thus we do not expect non-uniqueness in the solutions here (the function which represents a single ReLU neuron with minimum representation cost is simply a ReLU neuron). The purpose of this experiment was to emphasize the difference between single-task solutions and multi-task solutions in a more extreme setting and in higher dimensions.
> > >
> > > > In this example, the multi-task solution does not seem to be desirable but instead further away from the piecewise linear interpolant of the data than the single task solution.
> > >
> > > Our paper does not claim that multi-task solutions are **always** desirable. The main message is that they are different. We have clarified our use of the phrase “generalizes better” in the general response.
> > >
> > > > It also remains unclear how task alignment or high dimensionality impacts the conclusions drawn in the paper.
> > >
> > > The new experiments are in higher dimensions ($\mathbb{R}^5$). We agree that exploring how task alignment affects these solutions would be interesting, but our main message is that multi-task training in ReLU neural networks can be vastly different than single-task training **even with very little assumptions on the tasks**.

---

> > > > ### Author Response · Authors · 2024-08-12
> > > > **references (Follow-up to reviewer XF9Q)**
> > > >
> > > > [1] Ongie, Greg, et al. "A Function Space View of Bounded Norm Infinite Width ReLU Nets: The Multivariate Case" ICLR 2020
> > > >
> > > > [2] Zeno, Chen, et al. "How do minimum-norm shallow denoisers look in function space?." NeurIPS 2024.
> > > >
> > > > [3] Pilanci, Mert, and Tolga Ergen. "Neural networks are convex regularizers: Exact polynomial-time convex optimization formulations for two-layer networks." ICML 2020.
> > > >
> > > > [4] Hanin, Boris. "On The Implicit Bias of Weight Decay in Shallow Univariate ReLU Networks."

---

> > > > > ### Comment · Reviewer_XF9Q · 2024-08-13
> > > > >
> > > > > I thank the authors for these clarifications. After also reading the discussions with other reviewers, several of my concerns have been alleviated, in particular with the proof. I have slightly adjusted my score accordingly. I encourage the authors to further investigate how truly high dimension and task interaction affect the multi-task solution.

---

### Official Review · Reviewer_Snxf · 2024-07-06

**Soundness:** 3
**Presentation:** 3
**Contribution:** 3
**Rating:** 3
**Confidence:** 5

**Summary:**

The authors investigate the properties of solutions to multi-task shallow ReLU neural network training problems. It reveals a novel connection between neural networks and kernel methods, particularly focusing on interpolating training data while minimizing the sum of squared weights in the network. The findings highlight that while single-task solutions are non-unique and can exhibit undesirable behaviors, multi-task training leads to unique solutions with desirable generalization properties.

**Strengths:**

The authors offer a robust theoretical framework, providing new insights into how multi-task training in neural networks can lead to fundamentally different solutions compared to single-task training. This framework connects multi-task neural network solutions to kernel methods. It rigorously proves that under certain conditions, the solutions are unique and analogous to those in an RKHS.

**Weaknesses:**

1. The theorems show the difference between the single and multi-task solutions. However, the author's argument makes it hard to determine the difference between the 1-task and 2-task solutions.

2. Although [1] used the neural tangent kernel (NTK) framework, different from the idea in this paper, [1] showed that overparameterized neural networks on $\mathbb{R}$ is linear interpolation on the single task and it seems to be a generalization solution in the author's argument. This challenges the main contribution of this paper.

3. Under the RKHS framework (kernel methods), the author should quantify the generalization ability, like the generalization error bound, if they want to claim "the unique multi-task solution has desirable generalization properties that the single-task solutions can lack".

4. For the multivariate case, the exact nature of the kernel corresponding to the neural network solutions is not fully characterized.

If the authors can solve the above problems, I will consider raising the score.

[1] Generalization Ability of Wide Neural Networks on \mathbb{R}

**Questions:**

1. In Equation (3), do you mean $f_{\theta}(x_i)=y_i$ instead of $f_{\theta}(x_i)=x_i$?

2. What happens when $x$ is different in different tasks? Do your conclusions still hold?

**Limitations:**

The study is limited to shallow ReLU networks. Extending these results to other activation functions and deeper network architectures remains an open question.

---

> ### Author Rebuttal · Authors · 2024-08-06
>
> We thank the reviewer for their feedback, but it seems the reviewer has a **misunderstanding** about the results of our paper. Below, we address each of the comments individually.
> 1. **[1-task vs. 2-task solution]** Our result does indeed demonstrate the difference in solution between learning 1 task individually versus 2 tasks jointly. The description in Theorem 3.2 of conditions under which multi-task solutions are non-unique holds for any T > 1.
> 2. **[Relation to NTK]** Our results are based on established exact characterizations of the global minimizers of training neural networks with weight decay ([2], [1], [3], [4], [5], [6]). The global minimizers are non-unique in general, and in the univariate case, it is well known that the connect-the-dots solution is just one of (potentially infinitely) many solutions. **The NTK regime is a different, approximate analytical framework that relies on a number of assumptions which we do not employ in this work**. Namely, our analysis does not require that the learning rate be extremely small ("lazy training") or that the network have infinite width, both of which are necessary for the NTK regime and it is now well established that such assumptions are insufficient for explaining the success of neural networks ([8],[9],[10]]). Additionally, **the paper [7] cited by the reviewer does not address connections between neural kernel regime(s) and multi-task training/"task augmentation," which is one of the main novel contributions of our paper**. Finally, the conclusions about the results in [7] seem to be incorrect, as **[7] states that the solutions are *almost* linear interpolation under additional assumptions on the data**. In contrast our result states that the solutions are **exactly** connect-the-dots linear interpolation with very weak assumptions on the data which are almost always satisfied.
> 3. **[Generalization]** See the general response to all reviewers above. When we referenced generalization, we specifically meant to reference the "tempered overfitting" result in [3], which is specific to the univariate case.
> 4. **[Multi-variate case]** The kernel will depend on the specific neurons learned in training, as indicated in the paper. These will depend on the training data collectively across all tasks and the random initialization of the training process. However, the function learned for each task will be a standard kernel solution in terms of this kernel. In contrast, training separate networks for each task will, in general, produce different solutions that cannot be viewed as kernel solutions. Specifically, the solutions to single-task training with weight decay effectively minimize the $\ell_1$ norm of the output weights, as discussed in the paper and many of the referenced prior works.
>
> Regarding Question #2, this is an interesting setting, but our main focus in this paper is the standard multi-task learning scenario, in which all tasks share the same data points x.
>
> [1] Joshi, Nirmit, Gal Vardi, and Nathan Srebro. "Noisy interpolation learning with shallow univariate ReLU networks." ICLR 2024.
>
> [2] Hanin, Boris. "On The Implicit Bias of Weight Decay in Shallow Univariate ReLU Networks."
>
> [3] Ongie, Greg, et al. "A Function Space View of Bounded Norm Infinite Width ReLU Nets: The Multivariate Case." ICLR 2020.
>
> [4] Parhi, Rahul, and Robert D. Nowak. "What kinds of functions do deep neural networks learn? Insights from variational spline theory." SIAM Journal on Mathematics of Data Science 2022.
>
> [5] Savarese, Pedro, et al. "How do infinite width bounded norm networks look in function space?." COLT 2019.
>
> [6] Shenouda, Joseph, et al. "Variation Spaces for Multi-Output Neural Networks: Insights on Multi-Task Learning and Network Compression." JMLR 2024.
>
> [7] Lai, Jianfa, et al. "Generalization ability of wide neural networks on $\mathbb{R}$."
>
> [8] Jacot, Arthur, et al. "Feature Learning in $ L_2 $-regularized DNNs: Attraction/Repulsion and Sparsity." NeurIPS 2022
>
> [9] Arora, Sanjeev, et al. "On exact computation with an infinitely wide neural net." NeurIPS 2019.
>
> [10] Damian, Alex, et al. "Neural networks can learn representations with gradient descent." COLT 2022.

---

> > ### Comment · Reviewer_Snxf · 2024-08-09
> > **Thank you for your responds**
> >
> > 1. Are you referring to 'Lemma 3.2'? While you've illustrated the difference, the explanation lacks a clear description of the nature of this difference. For instance, what distinguishes a single-task classification problem where \( y_i \in \{0,1\} \) from a 'two-task' classification problem where $ y_i = (y_{i,1}, y_{i,2}) $ and $ y_{i,1} = 1$ if $y_{i,2} = 0 $? Alternatively, consider a scenario where a second task is created with random labels. What would be the implications for the original task?
> >
> > 2 & 3. I referenced [7] because it demonstrates that linear interpolation does not generalize well when the data contains noise. Are you suggesting that linear interpolation can indeed generalize under such conditions?

---

> > > ### Author Response · Authors · 2024-08-10
> > > **Follow-up to reviewer Snxf**
> > >
> > > We thank the reviewer for following up on this! Apologies, we meant to refer to Theorem 3.1 (there is no Theorem 3.2). The statement of **Theorem 3.1** precisely describes the conditions on the data and labels under which the unique solution for each task is connect-the-dots for any number of tasks $T > 1$, including $T=2$. For any dataset with two tasks, the solution is non-unique if and only if the conditions of Theorem 3.1 are satisfied, **otherwise the solution is unique and it is the connect-the-dots solution for both tasks (i.e. the solution is of the form of the right plot on Fig. 1 or the bottom row of Fig. 4)**. This is the difference in the nature of the solution between single task and multi-task learning.
> > >
> > > As a simple example, consider the data points {0, 1, 2, 3} and first task labels {0, 1, 1, 0}. A neural network trained on this task alone can interpolate the data either by a "peak" or a "plateau" (connect-the-dots) function with the same representational cost (similar to the example in Fig. 1). However, if we do multi-task learning with a second task with labels {0, 1, 0, 1}, then by Theorem 3.1 the neural network learns a unique solution which will be the connect-the-dots interpolant for both tasks, eliminating the possible "peak" solution for task 1 when learned by itself.
> > >
> > > It is possible to construct example sets of data and labels which admit multiple solutions by Theorem 3.1, but if we assume the task labels are real valued (as we do throughout the paper), i.e. a standard regression setting, the set of all such label sets has Lebesgue measure zero (see proof of Corollary 1 in Appendix 6.2). In other words, for real-valued labels, labels which admit non-unique solutions are exceedingly rare. A direct corollary of this fact is that, as long as the labels are sampled i.i.d. from a distribution which is absolutely continuous with respect to the Lebesgue measure on $\mathbb{R}^T$, the multi-task solution is connect-the-dots with probability 1. If the labels are binary-valued, then non-uniquess of solutions may not be as rare as in the real-valued case. For example, consider the two task setting above with task labels {0,1,1,0} and {1,0,0,1}. In this case, there can be multiple solutions to the multi-task training problem. However, our focus in this paper is on the regression setting, and in either case, Theorem 3.1 precisely describes the conditions under which solutions are and are not unique. **Please let us know if this explanation of the difference between one-task and two-tasks solution is still unclear.**
> > >
> > > Regarding the question about generalization: [7] shows that, for uniformly-spaced data points the generalization error of ("almost") connect-the-dots interpolation is *lower-bounded* by a constant with probability 1 as the number of samples $N$ grows large. In contrast, [1] (the paper we reference when referring to "desirable generalization properties" of connect-the-dots interpolation) shows that, for data points sampled from an absolutely continuous distribution, the generalization error of (exact) connect-the-dots interpolation is *upper-bounded* by a constant with probability 1 as $N$ grows large. Setting aside the different assumptions employed in both works, these two statements are not incompatible with each other. By the taxonomy in [9], the result in [1] shows that the connect-the-dots solution exhibits "tempered overfitting" which is worse than "benign overfitting" but not as terrible as "catastrophic overfitting".  We note however that **the main focus of the paper is to demonstrate the significant difference between single-task and multi-task solutions** and we will adjust our language about generalization benefits in the camera-ready version as described in the General Response.
> > >
> > > [1] Joshi, Nirmit, Gal Vardi, and Nathan Srebro. "Noisy interpolation learning with shallow univariate ReLU networks." ICLR 2024.
> > >
> > > [9] Mallinar, Neil, et al. "Benign, tempered, or catastrophic: Toward a refined taxonomy of overfitting." NeurIPS 2022.

---

> > > > ### Author Response · Authors · 2024-08-13
> > > > **Request for follow-up from reviewer Snxf**
> > > >
> > > > As the discussion period is coming to a close, we ask that the reviewer please let us know if we have clarified their concerns, and to update their score accordingly.

---

### Official Review · Reviewer_WQUb · 2024-07-15

**Soundness:** 3
**Presentation:** 3
**Contribution:** 2
**Rating:** 6
**Confidence:** 3

**Summary:**

Using the piece-wise linear data interpolation problem, the authors in this paper studied the solution obtained in single task learning and that in multi-task learning where interpolation functions are jointly obtained for multiple problems. Both numerical and empirical results are provided to show that muti-task learning high likely leads to unique optimal solution with different initializations, while the solution resulted from single-task learning is very sensitive to initialization. In the paper, the analysis is done first for the case where the input is univariate; and it is latter extended to the case of multivariate.

**Strengths:**

* The analysis is sound
* The conclusions are interesting, providing another prospective of seeing the advantage of multi-task learning over single task learning.

**Weaknesses:**

* There is a typo in Eq. (3). There is f_{\theta}(x_i)=x_i, the latter x_i should be y_i.
* The analysis is done based on a toy problem that is very simple, requiring technically, only two layers of neurons with relu activation in the first layer and identity activation in the second. However, networks for practical applications are much more complicated. In my own experience, very different solutions are clearly observed while training deep neural networks in multi-task learning setting with different random initializations. These observations do not align with the conclusions in this paper. I guess the underlying causes can be complicated, involving multiple factors, for example, as the authors pointed out whether global optimal solution is reached during each training may be in question. My concern is that given large gap between the toy setting in this paper and practical applications of deep neural networks, the real impact of this work or its significance is quite uncertain.

**Questions:**

None

---

> ### Author Rebuttal · Authors · 2024-08-06
>
> We thank the reviewer for their review and positive score. Our analysis does focus on two-layer networks, but the conclusions of the multivariate analysis can be used to reason about the behavior of functions learned at layers within a deeper network.  A ReLU layer in a deep network can be viewed as a shallow network (with inputs coming from the previous layer and outputs going into the next layer).  If the outputs of this layer are sufficiently diverse, then the behavior can be similar to the kernel-like behavior indicated by our analysis.  We have now also included new experiments in higher dimensions beyond the Fig. 5 example (see the general response above).
>
> Also, we agree that when training neural networks in multi-task settings, the solutions will not be exactly the same across different initializations, but this is not at odds with our results. In our setting, the input weights will depend both on the training data (collectively across all tasks) and the random initializations, but whatever input weights (and hence neurons) are learned, the output weights for each task are effectively minimizing a common weighted ridge regularization objective. We will update the language of the paper to make this point more clear.

---

> > ### Comment · Reviewer_WQUb · 2024-08-12
> >
> > I appreciate the authors to respond to my comments. After reading through review comments from other reviewers and authors' response, I would like to keep my initial assessment of this paper.

---

### Author Rebuttal · Authors · 2024-08-06

## General response to reviewers
We thank the reviewers for their helpful feedback and careful review as well as the AC.
Most reviewers agreed that our results are novel and provide new insights on multi-task learning with neural networks. In particular,
- Reviewer XF9Q noted that our main result is "to the best of my knowledge **novel** and **interesting**"
- Reviewer aewJ highlighted that our result "is arguably **unprecedented in the literature**" and "is an important step towards explaining why multi-tasking can lead to more robust models."

Here we provide a rebuttal to some of the common concerns across reviews.
___
### Clarified Claims on Generalization
Multiple reviewers raised concerns about the statement in our abstract that "the unique multi-task solution has desirable generalization properties that the single-task solutions can lack." We will remove this sentence from the abstract, as it does not describe our paper's main contribution(s), and generalization is not our main focus. **The main focus of the paper is to demonstrate the significant difference between single-task and multi-task solutions**, even when the tasks are statistically independent, and to highlight a novel connection between multi-task neural network training and kernel methods. Our intention in mentioning the "desirable generalization properties" was specific to the univariate case, in which the connect-the-dots univariate interpolant is known to exhibit "tempered overfitting" ([1]). Informally, this means that the connect-the-dots solution generalizes reasonably well, as opposed to other minimum norm solutions that can generalize arbitrarily poorly (see [1] for technical details of this result). We will update our language to clarify that this is the "desirable generalization property" being referenced, and that this is not a contribution or focus of our paper, merely a reference to an existing result which motivates why our work is interesting.

Several reviewers also noted that there is a typo in Equation 3: the correct expression should be $f_{\theta}(x_i) = y_i$, not $f_{\theta}(x_i) = x_i$.

### New Experimental Results
___
We have now included another experiment for the multi-variate setting that is higher dimensional and does not rely on the symmetric structure of the datapoints.

In this experiment we generate $T=25$ random teacher networks with one random ReLU neuron. We then construct a multi-task dataset by evaluating the teacher networks on the same $n=25$ random samples $x_i \in \mathbb{R}^5$ sampled i.i.d from a Gaussian distribution. Next, we train 25 single-output student networks on each task individually with $K=200$ neurons and one multi-task network on all tasks simultaneously.

The single-output networks learn very sparse solutions consisting of only 5-10 neurons for each task. In constrast the multi-task network consists of 155 active neurons and each active neuron contributes to all of the outputs of the network.
We plot the difference in the solutions obtained from training a student network on task $t$ independently of the rest vs. the $t^{\text{th}}$ output of the multi-task network along different directions in $\mathbb{R}^{5}$. We observe that the single-output student network trained on task $t$ learned the ground truth ReLU function and is a ReLU function in all directions. In contrast, for certain directions the $t^{\text{th}}$ output of the multi-task student network is very different from the single-output student network and is clearly not a ReLU function. This again supports our claim that the functions learned for each task via a multi-task neural networks can be very different from the function learned by a neural net trained on each task individually. Please see Fig. 1,  Fig. 2 and Fig. 3 in the attached PDF for the results.


[1] Joshi, Nirmit, Gal Vardi, and Nathan Srebro. "Noisy interpolation learning with shallow univariate ReLU networks." ICLR 2024.

---

### Decision · Program_Chairs · 2024-09-25

**Decision:**

Accept (poster)

**Comment:**

This paper examines the properties of solutions to multi-task shallow ReLU neural network training problems. This paper focuses on the fact that solutions to multi-task problems can align with those of kernel methods, uncovering a novel link between neural networks and kernel methods. In particular the paper considers the problem of interpolating training data while minimizing the sum of squared weights in the network. It is known that functions resulting from such single-task training problems solve a minimum norm interpolation problem and that these solutions are generally non-unique. However, the multiple solutions can differ significantly and may display undesirable behaviors, such as making large deviations from the training data. In contrast, the authors claim that solutions to multi-task training problems are different. In univariate settings, the authors prove that the solution to multi-task training is almost always unique, and each individual task’s solution is equivalent to minimum norm interpolation in a Sobolev space. Additionally, the authors purport to show that a unique multi-task solution exhibits favorable generalization properties that single-task solutions may lack. Finally, the authors provide empirical evidence and mathematical analysis, demonstrating that multivariate multi-task training can yield solutions related to minimum-norm interpolation in a Hilbert space.

The reviewers liked the fact that the model is nonlinear and the writing is clear. They raised various technical concerns that seems to have been addressed. One reviewer raised concerns about novelty. I disagree with this assessment and recommend acceptance.